# Atomically dispersed Cu coordinated Rh metallene arrays for simultaneously electrochemical aniline synthesis and biomass upgrading

Qiqi Mao[1], Xu Mu[1], Wenxin Wang[1], Kai Deng[1], Hongjie Yu[1], Ziqiang Wang[1], You Xu[1], Liang Wang[1] & Hongjing Wang [1] ✉

Organic electrocatalytic conversion is an essential pathway for the green conversion of low-cost organic compounds to high-value chemicals, which urgently demands the development of efficient electrocatalysts. Here, we report a Cu single-atom dispersed Rh metallene arrays on Cu foam for cathodic nitrobenzene electroreduction reaction and anodic methanol oxidation reaction. In the coupled electrocatalytic system, the $Cu_{single-atom}$-Rh metallene arrays on Cu foam requires only the low voltages of 1.18 V to reach current densities of 100 mA cm$^{-2}$ for generating aniline and formate, with up to ~100% of nitrobenzene conversion/ aniline selectivity and over ~90% of formate Faraday efficiency, achieving synthesis of high-value chemicals. Density functional theory calculations reveal the electron effect between Cu single-atom and Rh host and catalytic reaction mechanism. The synergistic catalytic effect and H*-spillover effect can improve catalytic reaction process and reduce energy barrier for reaction process, thus enhancing electrocatalytic reaction activity and target product selectivity.

The utilization of renewable energy is a crucial pathway for solving the increasing energy crisis and promoting the green low-carbon transformation of energy[1,2]. The organic electrocatalytic conversion, driven by electricity generated from renewable energy sources at ambient temperature and pressure[3,4], is a green synthesis route for achieving controlled conversion of low-cost organic compounds into high-value chemicals and holds considerable research significance and application potential in the chemical industry and organic synthesis[5–7]. Among the numerous organic electrocatalytic conversion reactions, the cathodic nitrobenzene (Ph-NO$_2$) electroreduction for aniline (Ph-NH$_2$) synthesis is regarded as a more low-carbon and environment-friendly green Ph-NH$_2$ synthesis process using H$_2$O and electrons as the hydrogen source and reductant, compared with high pollution and harsh conditions of traditional chemical synthesis method[8–11]. Notably,

the anodic reaction for the conventional cathodic Ph-NO$_2$ electroreduction reaction (Ph-NO$_2$ ERR) is a sluggish kinetic oxygen evolution reaction (OER)[12,13]. In contrast, the anodic biomass electrooxidation reaction not only can achieve the upgrading of cheap biomass but also offers the advantages of low energy consumption and high electrolysis efficiency[14–16]. Currently, the cheap and widespread methanol (CH$_3$OH, 350 USD/t) is considered an appropriate precursor for the high-value formate (HCOO$^-$, 1300 USD/t) synthesis via anodic electrooxidation reaction[17,18], and methanol electrooxidation reaction (MOR) coupled with various electrocatalytic reactions has attracted widespread research interest worldwide[19,20]. In view of this, the construction of Ph-NO$_2$ ERR-MOR coupled electrolytic system is a promising strategy for achieving high-value chemical synthesis from sustainable organic electrocatalytic conversion. Nevertheless, both Ph-NO$_2$-to-Ph-NH$_2$ and

[1]State Key Laboratory Breeding Base of Green-Chemical Synthesis Technology, College of Chemical Engineering, Zhejiang University of Technology, Hangzhou 310014, P. R. China. ✉e-mail: hjw@zjut.edu.cn

methanol-to-formate are multi-step reaction processes and are limited by numerous side reactions[21,22]. To address these bottlenecks, it is fundamental to explore efficient bifunctional electrocatalysts for activating reaction pathways of the directed synthesis towards Ph-NH$_2$ and formate.

Recently, the single-atom alloys (SAAs) catalysts, consisting of exogenous isolated metal atoms dispersed on the surface of the metal host[23,24], as a promising material with the advantages of high active atom utilization of single-atom catalysts (SACs) and alloy synergistic effect[25,26], have been in the field of various electrocatalytic energy conversions[27–30]. For example, Duan and co-workers designed a Ru$_1$Cu SAAs with isolated Ru atoms on Cu nanowires for the electrocatalytic conversion of 5-hydroxymethylfurfural (HMF) to 2,5-dihydroxymethylfuran (DHMF), where the introduction of Ru single-atom sites facilitates the dissociation of H$_2$O to produce H* species for the HMF hydrogenation process[31]. Thus, the electron effect and metal-support interaction of SAAs can promote effective dissociation of reactants and optimize the adsorption/desorption of key intermediates to achieve an optimum balance between reactants and intermediates, leading to high activity and selectivity[32,33]. Furthermore, the electrocatalytic activity of SAAs catalysts can be further improved by the precise regulation for morphology and structure[34,35]. Metallene, a group of graphene-like two-dimensional (2D) nanomaterials with a thickness less than 5 nm[36,37]. Due to their flexible microstructural tunability, highly exposed metal active sites as well as simple and well-defined structure model[38,39], metallene can serve as a desirable carrier for immobilizing single-atom, clusters, or nanoparticles, which has attracted great research interest[40,41]. For example, Xiaoqiang Cui et al. report a dispersed MoO$_x$ on Rh metallene (MoO$_x$-Rh metallene) for boosting alkaline HER[42]. Chu's group prepare a single-atom Bi alloyed Pd metallene (Bi$_1$Pd metallene) that shows excellent NO$_3^-$ electroreduction reaction activity and near 100% Faradaic efficiency of NH$_3$[43]. Therefore, the design and development of SAA metallene are extremely promising for improving electrocatalytic activity and catalytic product selectivity.

In this work, we report a synthesis of Cu single-atom dispersed Rh metallene arrays on Cu foam (Cu$_{SA}$-Rh MAs/CF) by a facile and rapid one-step solvothermal approach. As a bifunctional electrocatalyst, the Cu$_{SA}$-Rh MAs/CF displays superior electrocatalytic activity for Ph-NO$_2$ ERR and MOR. For the constructed Ph-NO$_2$ ERR-MOR coupled electrocatalytic system, the low voltage of only 1.05/1.18 V achieves current densities of 50/100 mA cm$^{-2}$ for efficient conversion of Ph-NO$_2$-to-Ph-NH$_2$ and methanol-to-formate, with Ph-NO$_2$ conversion and Ph-NH$_2$ selectivity up to ~100% and HCOO$^-$ FE reaching over ~90% on Cu$_{SA}$-Rh MAs/CF, which not only enables the simultaneous cathodic and anodic organic electrocatalytic conversion for the synthesis of high-value chemicals but also maximizes the energy efficiency. Moreover, density functional theory (DFT) calculations further reveal the synergistic catalysis effect and H*-spillover effect induced by the local electron change between the isolated Cu single-atom and the Rh host, which promotes the rapid conversion of the reactants to key intermediates and rapid desorption of the target products thus enhancing electrocatalytic reaction activity and targeted product selectivity.

## Results

### Synthesis and characterization of Cu$_{SA}$-Rh MAs/CF

The Cu$_{SA}$-Rh MAs/CF was synthesized using a straightforward one-step solvothermal method (Supplementary Fig. 1), which contains rhodium (II) acetate dimer (C$_8$H$_{12}$O$_8$Rh$_2$), N,N-dimethylformamide (DMF), potassium hydroxide (KOH), ethylene glycol (EG), diethylenetriamine (DETA) and CF as reactants. In this synthesis process, the formed DMA by the synergistic interaction of DMF and KOH can induce the 2D nanosheet structure growth by facet control as well as diamine ligand DETA can chelate with metal ions to reduce the metal reduction rate thus promoting the formation of 2D nanosheet structure[44].

Supplementary Fig. 2 shows that the surface color of formed Cu$_{SA}$-Rh MAs/CF is black, which is significantly different from that of CF. Figure 1a displays uniform ultrathin nanosheets grown on the CF surface to form metallene arrays, and the nanosheets are tightly interconnected to each other to form a security wall-like structure, which contributes to providing enough active sites as well as facilitating rapid charge transfer and mass transfer during the electrocatalysis process[45]. Transmission electron microscopy (TEM) images of Cu$_{SA}$-Rh MAs further display the ultrathin nanosheet-like metallene structure with lateral dimensions of around several hundred nanometers (Fig. 1b). The thickness of a single nanosheet in Cu$_{SA}$-Rh MAs was measured to be ~1.51 nm corresponding to 7-8 atomic layers (Fig. 1c and Supplementary Fig. 3), further proving the ultrathin nature for Cu$_{SA}$-Rh MAs. The ultrathin 2D nanosheet structure with certain curvature can provide highly accessible surface atoms, highly exposed active sites, and rich defect structures, which is conducive to promoting the electrocatalytic process[37]. Figure 1d presents the aberration-corrected high-angle annular dark field scanning transmission electron microscopy (AC-HAADF-STEM) and corresponding elemental mapping images of Cu$_{SA}$-Rh MAs, revealing the homogeneous distribution of Cu atoms in Cu$_{SA}$-Rh MAs. The Rh/Cu atomic ratio was further determined to be approximately 93.6/6.4 via the TEM energy dispersive X-ray spectroscopy (TEM-EDS, Supplementary Fig. 4). The mass ratio (Rh/Cu = 95.1/4.9) and atomic ratio (Rh/Cu = 92.6/7.4) of Rh/Cu in Cu$_{SA}$-Rh MAs were further analyzed by inductively coupled plasma optical emission spectroscopy (ICP-OES) (Supplementary Fig. 5), which is close to the results obtained from TEM-EDS. Furthermore, as revealed by the electron energy loss spectrum (EELS) of Cu$_{SA}$-Rh MAs (Supplementary Fig. 6), the energy loss peak around 498.8 eV can be assigned to the Rh M electron transition (Supplementary Fig. 6a)[46,47], and the energy loss peak around 933.1 eV in Supplementary Fig. 6b can be assigned to the Cu L electron transition[48]. The EELS data of Cu$_{SA}$-Rh MAs further reveal the existence of Rh and Cu elements. The AC-HAADF-STEM images of Cu$_{SA}$-Rh MAs were collected and analyzed for further investigating its structure at the atomic scale. As shown in Fig. 1e–f, the obvious vacancy defects are observed on the basal surface of Cu$_{SA}$-Rh MAs and the presence of atomic vacancies is also verified by the corresponding integrated pixel intensity profile in the red region of Fig. 1e. Moreover, the lattice spacing ($d = 0.220$ nm) of Cu$_{SA}$-Rh MAs can be indexed to the typical face-centered cube (fcc) Rh (111) facet and the appearance of amorphous sites in Cu$_{SA}$-Rh MAs reveals the presence of crystalline and amorphous phases (Fig. 1g). In the X-ray diffraction (XRD) pattern (Fig. 1h), the characteristic peaks of Cu$_{SA}$-Rh MAs can be assigned to a typical fcc metallic Rh phase (No. 05−0685). Notably, the characteristic peaks of Cu$_{SA}$-Rh MAs exhibit a negative shift compared with the Rh JCPDS card, originating from the curved 2D geometrical structure and the introduction of Cu atoms[38]. Moreover, the poor crystallinity is revealed by the weak and broad characteristic peaks of Cu$_{SA}$-Rh MAs in consistent with the selected area electron diffraction (SAED, inset in Fig. 1h) result, which further indicates the presence of the crystalline and amorphous phases in Cu$_{SA}$-Rh MAs. In detail, the vacancy defects and amorphous sites as highly active low-coordination sites can break the inherent crystal arrangement to cause atomic unsaturated bonding and readjust the local electron structure for optimizing the surface electron structure of the electrocatalyst, which is beneficial for improving the electrocatalytic activity[37]. The Cu single-atom was analyzed by AC-HAADF-STEM image and 3D topographic atom images. Figure 2a shows that some individual dark dots (Cu atoms) can be observed on the surface of Cu$_{SA}$-Rh MAs owing to the lower atomic number of Cu (29) compared with Rh (45). The low-intensity dots also indicate the dispersion situation of the isolated Cu atoms on the surface of Cu$_{SA}$-Rh MAs (Fig. 2b, c). The corresponding integrated pixel intensity profile also illustrates the isolated low-intensity Cu atoms dispersed surrounding the high-intensity Rh atoms on the crystal surface (Fig. 2d), further proving the presence of isolated Cu

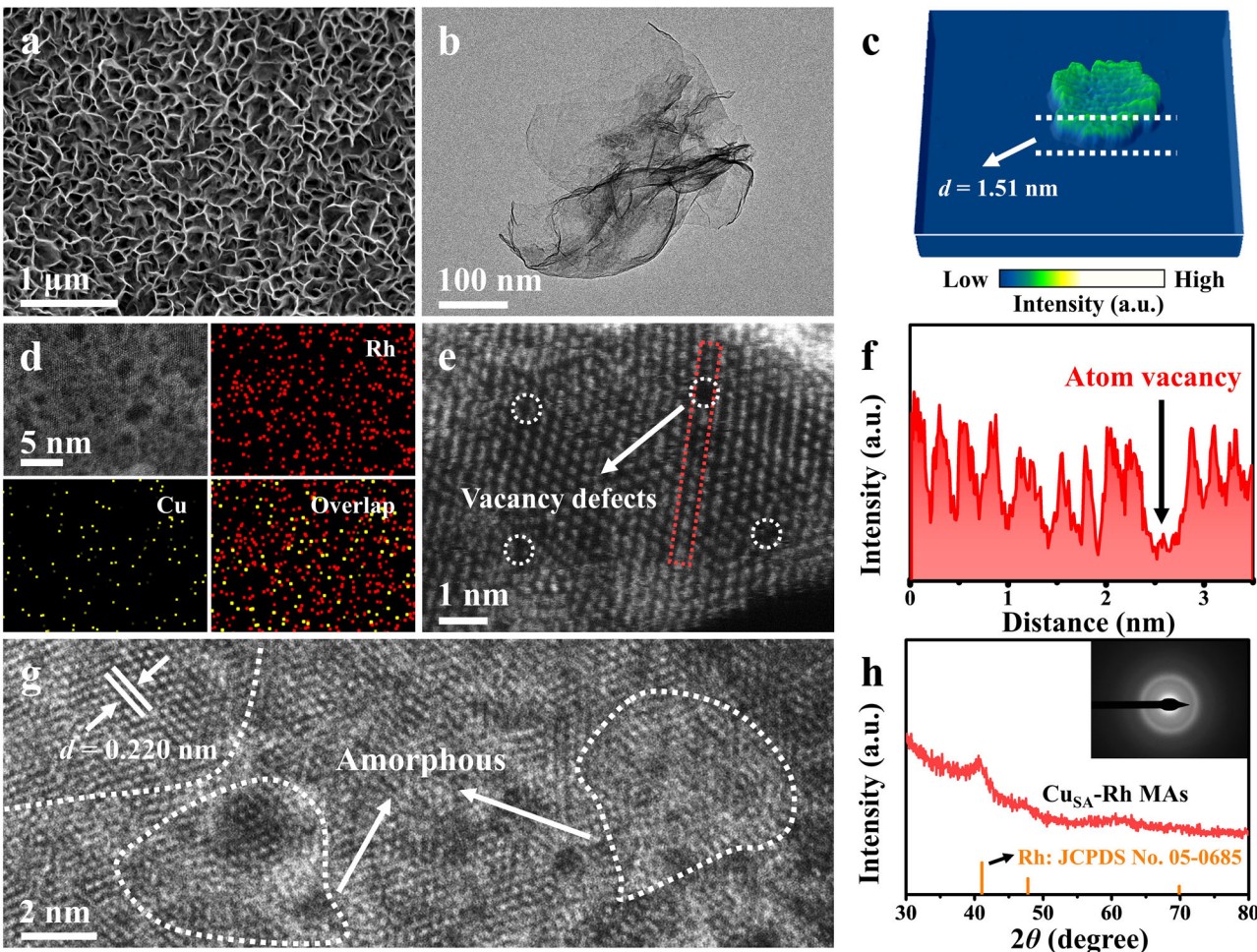

**Fig. 1 | Morphological and structure characterization of Cu$_{SA}$-Rh MAs/CF. a** SEM image of Cu$_{SA}$-Rh MAs/CF. **b** TEM image, **c** 3D view for the corresponding AFM image, **d** AC-HAADF-STEM image, and the corresponding elemental mapping images of Cu$_{SA}$-Rh MAs. **e** AC-HAADF-STEM image of Cu$_{SA}$-Rh MAs. **f** The integrated pixel intensity profile for the selected red region in **e**. **g** AC-HAADF-STEM image of Cu$_{SA}$-Rh MAs. **h** XRD pattern of Cu$_{SA}$-Rh MAs and the inset in **h** displays the corresponding SAED pattern of Cu$_{SA}$-Rh MAs.

single-atom on the Cu$_{SA}$-Rh MAs. The strong metallic interaction between the isolated Cu single-atom and Rh metallene support can modulate the electronic structure of active sites and form the activated Rh-coordinated Cu single-atom active sites, which is beneficial for optimizing the adsorption and activation between active sites and reactants during electrocatalytic reactions[25,43].

The elemental composition, chemical valence, and coordination environment for Cu$_{SA}$-Rh MAs were investigated through the X-ray photoelectron spectroscopy (XPS), X-ray absorption near-edge structure (XANES), and extended X-ray absorption fine structure (EXAFS) tests. As shown in Fig. 2e, the Rh 3$d$ spectrum of Cu$_{SA}$-Rh MAs reveals that four obvious characteristic peaks (307.42 eV, 308.75 eV, 312.19 eV, 313.98 eV) are ascribed to Rh$^0$ 3$d_{5/2}$, Rh$^{3+}$ 3$d_{5/2}$, Rh$^0$ 3$d_{3/2}$ and Rh$^{3+}$ 3$d_{3/2}$, respectively[46,49], indicating the presence of Rh in Cu$_{SA}$-Rh MAs primarily as the metallic state. In the Cu 2$p$ spectrum of Cu$_{SA}$-Rh MAs (Fig. 2f), two obvious characteristic peaks are at 932.50 eV and 952.37 eV ascribed to Cu$^0$[50], and other characteristic peaks at 934.76 eV, 954. 49 eV and 942.21 eV can be ascribed to Cu$^{2+}$ and satellite peak[51]. As observed from Rh $K$-edge XANES, EXAFS, and wavelet transform (WT) spectra (Fig. 2g–i), the similar features of Cu$_{SA}$-Rh MAs with Rh foil reveal that the surface valence and coordination structure of Rh is not significantly changed by the introduction of Cu single-atom. Notably, based on the WT spectra analysis for Cu$_{SA}$-Rh MAs and Rh foil (Fig. 2i), the Rh-Rh/Cu-Rh intensity maximum of Cu$_{SA}$-Rh MAs exhibits a negative shift of -0.33 Å$^{-1}$ compared with Rh-Rh

intensity maximum of Rh foil, which is induced by the coordination of the Cu-Rh bond. Supplementary Figs. 7–9 shows the Rh $K$-edge experimental and fitting Fourier-transformed EXAFS spectra for Cu$_{SA}$-Rh MAs, Rh foil, and Rh$_2$O$_3$, indicating a good fitting result. The Cu $K$-edge XANES spectra indicate that the absorption edge of Cu$_{SA}$-Rh MAs is positioned between Cu foil and CuO (Fig. 2j), indicating an increase in the valence state of Rh for Cu$_{SA}$-Rh MAs owing to the electron transfer from Cu to Rh. This phenomenon reveals the electronic interaction between Cu single-atom and Rh in Cu$_{SA}$-Rh MAs. The Cu $K$-edge EXAFS spectra show the distinct peak of Cu$_{SA}$-Rh MAs at 2.43 Å ascribed to the Cu-Rh bond, obviously distinct with Cu-Cu (2.23 Å) and CuO (1.51 Å) bands (Fig. 2k), revealing the presence of dispersed Cu single-atom. The atomically distributed Cu atoms were also identified based on the fitting results (Supplementary Figs. 10–12 and Supplementary Table 1). As displayed in Fig. 2l, The WT spectra for Cu$_{SA}$-Rh MAs, Rh foil, and Rh$_2$O$_3$ further indicate the presence of isolated Cu single-atom, where the intensity maximum (-10.31 Å$^{-1}$) of Cu$_{SA}$-Rh MAs is ascribed to the Cu-Rh bond compared with Cu-Cu (-8.01 Å$^{-1}$) and CuO (-6.76 Å$^{-1}$) intensity maximums.

**Electrocatalytic performance for Ph-NO$_2$ ERR**
The cathodic electrochemical Ph-NH$_2$ synthesis activity of Cu$_{SA}$-Rh MAs/CF was evaluated in an H-type electrolyzer separated via a Nafion membrane under ambient conditions, consisting of a working electrode (prepared electrocatalysts), a reference electrode (Hg/HgO

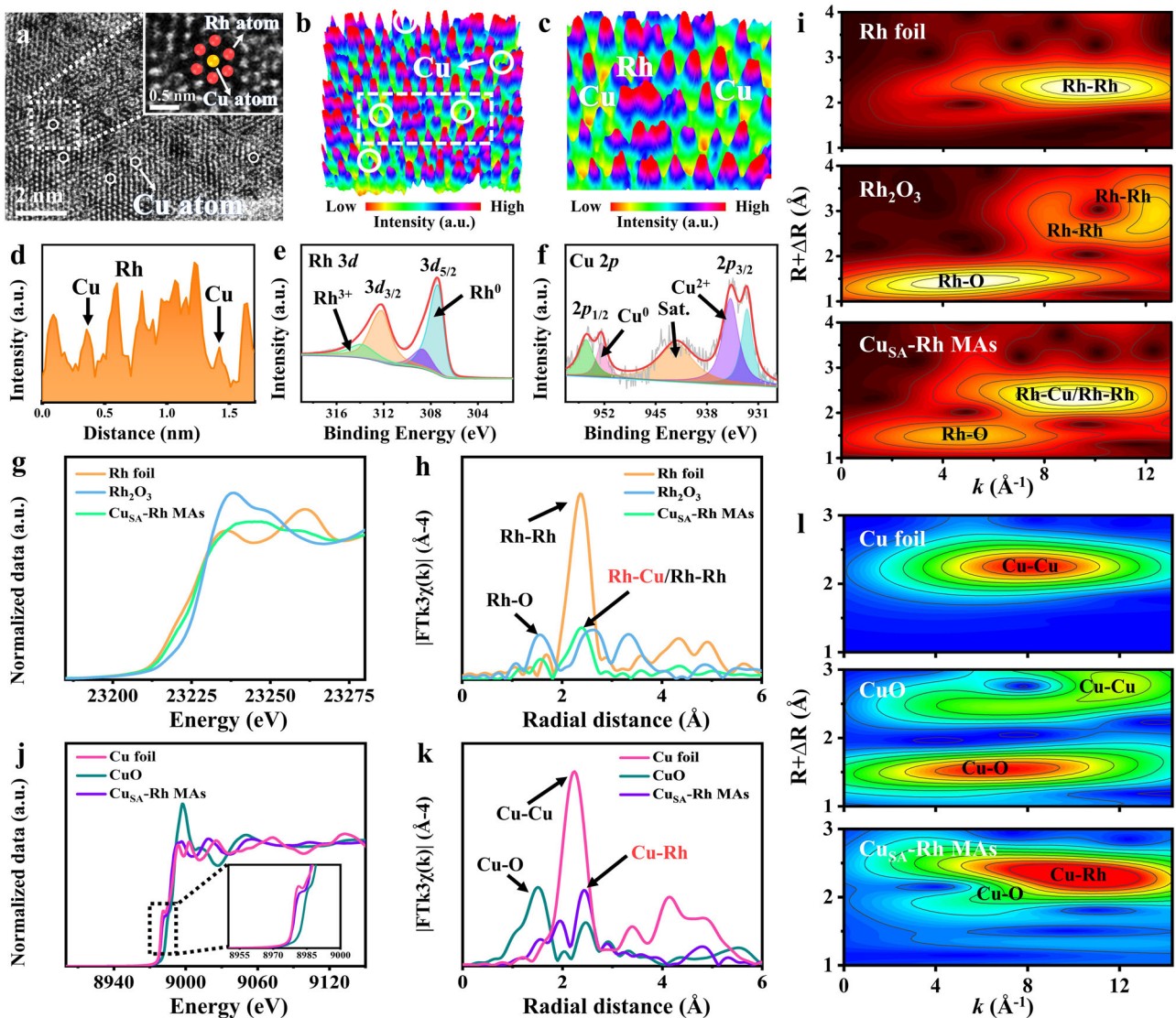

**Fig. 2 | Structure characterization of Cu$_{SA}$-Rh MAs/CF. a** AC-HAADF-STEM image of Cu$_{SA}$-Rh MAs and the inset in a displays the magnified image of the selected white region in **a**. **b, c** 3D topographic atom images of Cu$_{SA}$-Rh MAs. **d** The integrated pixel intensity profile for the selected region in **b**. **e** Rh 3$d$ XPS and **f** Cu 2$p$ XPS spectra of Cu$_{SA}$-Rh MAs. **g** Rh $K$-edge XANES spectra, **h** Fourier transformed EXAFS spectra and **i** EXAFS wavelet transform diagrams of Rh foil, Rh$_2$O$_3$, and Cu$_{SA}$-Rh MAs. **j** Cu $K$-edge XANES spectra, **k** Fourier transformed EXAFS spectra, and **l** EXAFS wavelet transform diagrams of Cu foil, CuO, and Cu$_{SA}$-Rh MAs.

electrode) and a counter electrode (Pt foil). As shown in Fig. 3a, linear sweep voltammetry (LSV) curves of Cu$_{SA}$-Rh MAs/CF were recorded in 1 M KOH and 1 M KOH + 5 mM Ph-NO$_2$ solutions. Apparently, a rapid increase in cathodic current density is observed at -0.4 V (vs RHE) in the presence of Ph-NO$_2$, suggesting the proceeding of Ph-NO$_2$ electroreduction on Cu$_{SA}$-Rh MAs/CF. Furthermore, at the applied potential range from −0.7 to 0.5 V (vs RHE), the significantly improved overall current density and lower onset potential in a 1 M KOH + 5 mM Ph-NO$_2$ solution further demonstrate the favorable Ph-NO$_2$ ERR activity on Cu$_{SA}$-Rh MAs/CF. To investigate the potential-dependent Ph-NO$_2$ conversion and Ph-NH$_2$ selectivity on Cu$_{SA}$-Rh MAs/CF, the chronoamperometric (*i-t*) measurements were performed at various applied potentials and the electrolyzed products were quantified by GC analysis. As shown in Fig. 3b, the Ph-NO$_2$ conversion of Cu$_{SA}$-Rh MAs/CF is as high as -100% at various applied potentials, revealing the superior Ph-NO$_2$ adsorption and electroreduction ability on Cu$_{SA}$-Rh MAs/CF. For Ph-NH$_2$ selectivity (Fig. 3c), the Ph-NH$_2$ selectivity of Cu$_{SA}$-Rh MAs/CF gradually increases from the applied potentials of 0.2 to −0.2 V (vs RHE), and the Ph-NH$_2$ selectivity is calculated to be -99.7% at −0.1 V (vs RHE) approaching 100%, which indicates the Ph-NO$_2$ can be

selectively electroreduced to Ph-NH$_2$ via hydrogenation using H$_2$O as the hydrogen source on Cu$_{SA}$-Rh MAs/CF. Notably, the low Ph-NH$_2$ selectivity of lower cathodic potentials can be ascribed to the weak ability of H$_2$O dissociation to produce active H*, and the production of active H* is gradually improved on Cu$_{SA}$-Rh MAs/CF with the increase of cathodic potentials thus enhancing the Ph-NH$_2$ selectivity[11]. Besides, Fig. 3d and Supplementary Fig. 13 reveal that the Ph-NO$_2$ can be completely converted to Ph-NH$_2$ as well as the high-boiling by-products are not present in the obtained electrolyte at an applied potential of −0.1 V (vs RHE), further indicating the superior activity for electrochemical Ph-NH$_2$ synthesis on Cu$_{SA}$-Rh MAs/CF. As revealed in Fig. 3e, f, the time-dependent conversion, and yield illustrate that the Ph-NH$_2$ concentration increases and the Ph-NO$_2$ concentration decrease over time, indicating a gradual and complete conversion of Ph-NO$_2$ to Ph-NH$_2$ on Cu$_{SA}$-Rh MAs/CF at an applied potential of −0.1 V (vs RHE) for 1 h. To further investigate the mechanism for the improved Ph-NO$_2$ ERR to Ph-NH$_2$ activity over Cu$_{SA}$-Rh MAs/CF, several comparison experiments were carried out. LSV curves for various electrocatalysts demonstrate that the Cu$_{SA}$-Rh MAs/CF possesses a stronger electroreduction activity compared with Rh metallene-CF

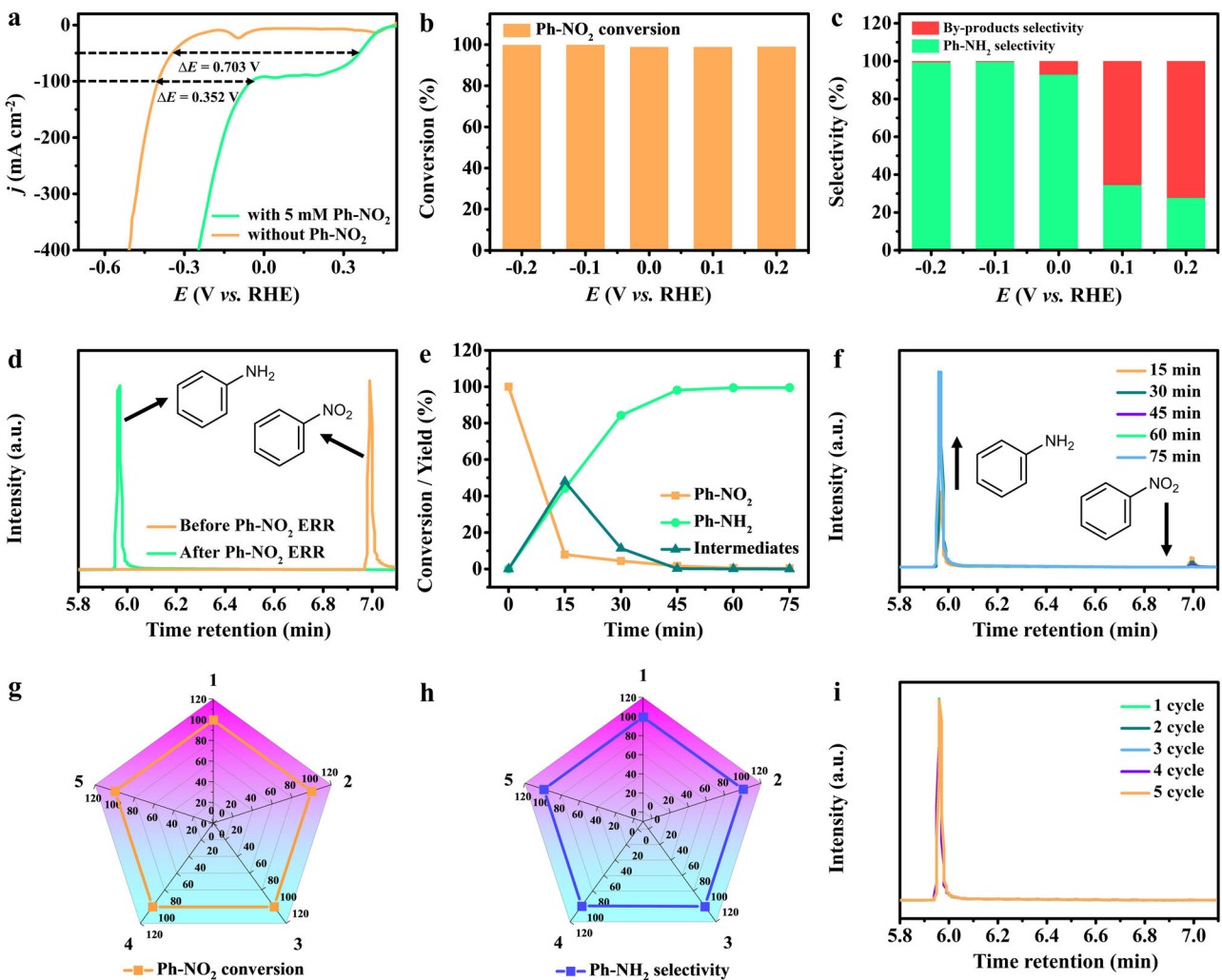

**Fig. 3 | Electrocatalytic Ph-NO$_2$ ERR performance. a** IR-corrected LSV curves for Cu$_{SA}$-Rh MAs/CF (catalyst loading: 2.5 mg cm$^{-2}$) in 1 M KOH (pH = 13.9) and 1 M KOH + 5 mM Ph-NO$_2$ (pH = 13.7) solutions. **b** Ph-NO$_2$ conversion and **c** Ph-NH$_2$ selectivity for Cu$_{SA}$-Rh MAs/CF at various applied potentials in 1 M KOH + 5 mM Ph-NO$_2$ (pH = 13.7) solutions. **d** GC results of products before and after Ph-NO$_2$ ERR for Cu$_{SA}$-Rh MAs/CF at −0.1 V (vs RHE). **e** Time-dependent conversion and yield for Ph-NO$_2$ ERR of Cu$_{SA}$-Rh MAs/CF. **f** GC results for Ph-NO$_2$ ERR of Cu$_{SA}$-Rh MAs/CF at −0.1 V (vs RHE) with various time. **g** Ph-NO$_2$ conversion, **h** Ph-NH$_2$ selectivity and **i** GC results for Ph-NO$_2$ ERR of Cu$_{SA}$-Rh MAs/CF at −0.1 V (vs RHE) measured for 5 successive cycles in 1 M KOH + 5 mM Ph-NO$_2$ (pH = 13.7) solutions.

(Rhene-CF), Rh nanoparticles-CF (Rh NPs-CF), and CF (Supplementary Fig. 14a), and the Ph-NH$_2$ selectivity (99.7%) of Cu$_{SA}$-Rh MAs/CF is much higher than those of Rhene-CF (64.7%), Rh NPs-CF (59.3%) and CF (28.3%) (Supplementary Fig. 14b). Moreover, with respect to the standard potential (0.89 V vs RHE) of the Ph-NO$_2$ ERR[21], the overpotential (0.532 V vs RHE) of Cu$_{SA}$-Rh MAs/CF is lower than those of Rhene-CF (0.667 V vs RHE), Rh NPs-CF (0.619 V vs RHE) and CF (0.638 V vs RHE) for achieving a current density of −50 mA cm$^{-2}$ (Supplementary Fig. 15), further suggesting a superior Ph-NO$_2$ ERR activity on Cu$_{SA}$-Rh MAs/CF. The improved activity of Ph-NO$_2$ ERR to Ph-NH$_2$ for Cu$_{SA}$-Rh MAs/CF originates from the ultrathin metallene array structure and the synergistic effect of isolated Cu single-atom with Rh host. Moreover, the stability of electrochemical Ph-NH$_2$ synthesis is also a critical parameter for evaluating Ph-NO$_2$ ERR activity. After 5 repeated cycles of testing on Cu$_{SA}$-Rh MAs/CF, the decay was almost negligible for Ph-NO$_2$ conversion, Ph-NH$_2$ selectivity, and Ph-NH$_2$ intensity (Fig. 3g–i), revealing a superb stability for Ph-NO$_2$ ERR to Ph-NH$_2$ of Cu$_{SA}$-Rh MAs/CF.

## Electrocatalytic performance for MOR

A single-chamber cell with a three-electrode system was utilized to investigate the anodic electrochemical methanol (CH$_3$OH)

upgrading for high value-added formate (HCOO$^-$) production on Cu$_{SA}$-Rh MAs/CF. The working electrode contained prepared electrocatalysts, while Hg/HgO electrode and carbon rod served as reference electrode and counter electrode. Figure 4a displays the LSV curves of Cu$_{SA}$-Rh MAs/CF recorded in 1 M KOH with different CH$_3$OH concentrations solutions. Visibly, the Cu$_{SA}$-Rh MAs/CF exhibits an optimal MOR activity when the CH$_3$OH concentration reaches 4 M. The oxidation potentials ($E_{50}$, $E_{100}$, $E_{150}$, and $E_{200}$) of only 1.40, 1.44, 1.46, and 1.47 V (vs RHE) are required for reaching the current densities of 50, 100, 150 and 200 mA cm$^{-2}$ at a 1 M KOH + 4 M CH$_3$OH solution (Fig. 4b). which is lower than those of other CH$_3$OH concentrations. Notably, the MOR activity decays significantly when the CH$_3$OH concentration increases to 8 M, which can originate from the poor conductivity of the mixed solution as well as excessive oxidation species adsorbed on the active site to hinder the reaction[20]. Hence, it is considered that a 4 M CH$_3$OH is appropriate for this system. Figure 4c illustrates the LSV curves of Cu$_{SA}$-Rh MAs/CF recorded in MOR (with 4 M CH$_3$OH) and OER (without 4 M CH$_3$OH). Observably, the onset potential is significantly reduced after the addition of CH$_3$OH. The oxidation potentials ($E_{50}$ = 1.40, $E_{100}$ = 1.44, $E_{150}$ = 1.46 and $E_{200}$ = 1.47 V vs RHE) for MOR are lower than those for OER ($E_{50}$ = 1.57, $E_{100}$ = 1.60, $E_{150}$ = 1.62 and $E_{200}$ = 1.64 V

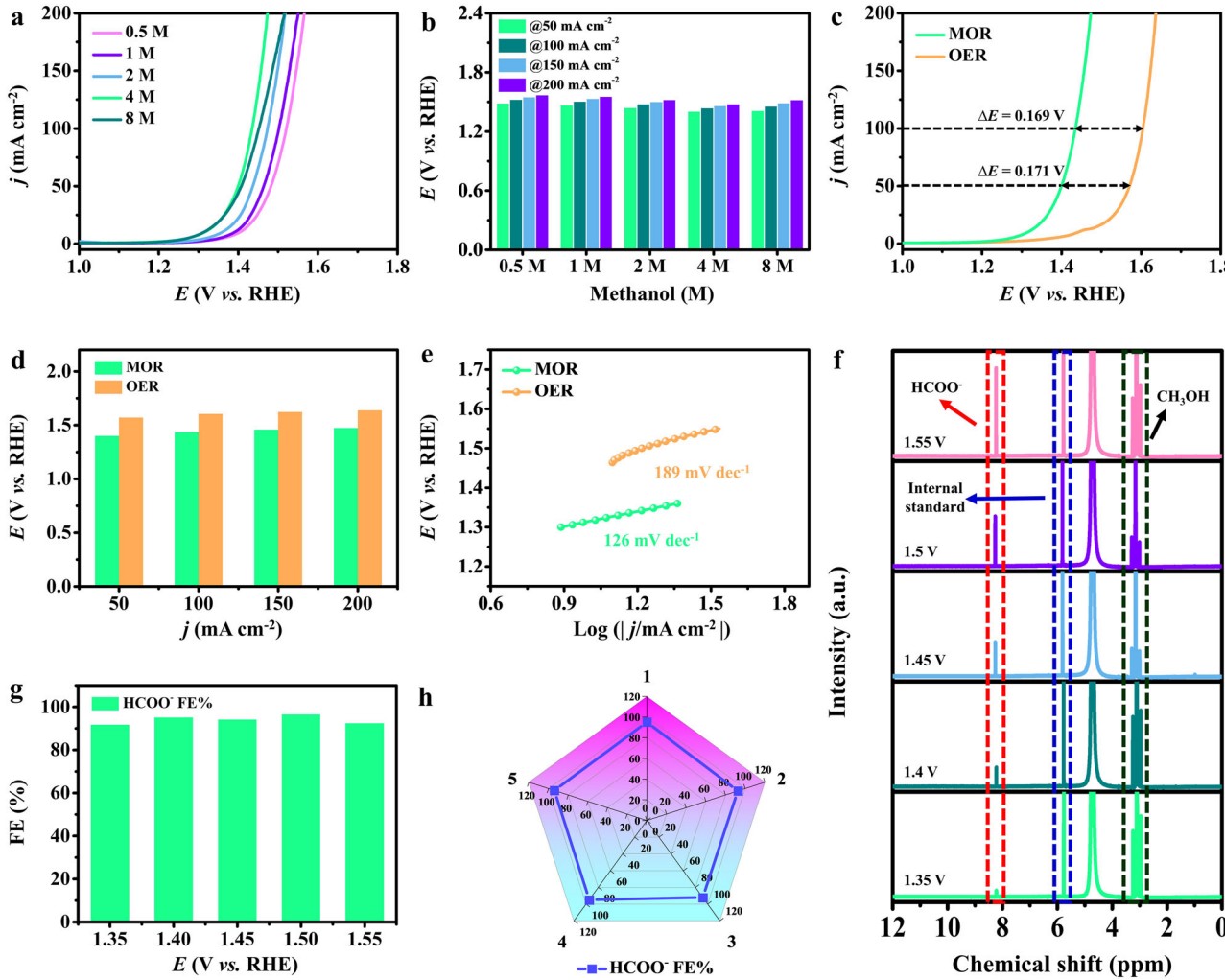

**Fig. 4 | Electrocatalytic MOR performance. a** IR-corrected LSV curves for Cu$_{SA}$-Rh MAs/CF (catalyst loading: 2.5 mg cm$^{-2}$) and **b** corresponding overpotentials comparison at 50/100/150/200 mA cm$^{-2}$ in 1 M KOH containing 0.5 M (pH = 13.9), 1 M (pH = 13.8), 2 M (pH = 13.8), 4 M (pH = 13.7) and 8 M (pH = 13.7) CH$_3$OH solutions. **c** LSV curves, **d** corresponding overpotentials comparison at 50/100/150/200 mA cm$^{-2}$ and **e** Tafel plots of Cu$_{SA}$-Rh MAs/CF for MOR and OER in 1 M KOH

(pH = 13.9) and 1 M KOH + 4 M CH$_3$OH (pH = 13.7) solutions. **f** $^1$H NMR spectra of products after MOR at various applied potentials. **g** HCOO$^-$ FEs of Cu$_{SA}$-Rh MAs/CF at various applied potentials in 1 M KOH + 4 M CH$_3$OH (pH = 13.7) solutions. **h** HCOO$^-$ FEs of Cu$_{SA}$-Rh MAs/CF at 1.4 V (vs RHE) measured for 5 successive cycles in 1 M KOH + 4 M CH$_3$OH (pH = 13.7) solutions.

vs RHE) and the Tafel slope value of MOR (126 mV dec$^{-1}$) is also smaller than that of OER (189 mV dec$^{-1}$) (Fig. 4d, e), indicating the thermodynamic favorability of MOR over OER owing to faster kinetics and smaller oxidation potential. To further investigate the products of the anodic MOR on Cu$_{SA}$-Rh MAs/CF, the *i-t* measurements were performed at various applied potentials and the electrolyzed products were analyzed via the nuclear magnetic resonance (NMR) spectra. As presented in Fig. 4f, the key value-added chemical HCOO$^-$ is produced during the MOR process, and the HCOO$^-$ concentration gradually increases with higher anodic applied potential. Figure 4g presents that the HCOO$^-$ Faradaic efficiencies (FEs) for Cu$_{SA}$-Rh MAs/CF reaches over 90% at various applied potentials (1.35−1.55 V vs RHE), where the HCOO$^-$ FE of Cu$_{SA}$-Rh MAs/CF is ~96.5% at 1.5 V (vs RHE), which reveals the excellent selectivity for the conversion of CH$_3$OH to HCOO$^-$ on Cu$_{SA}$-Rh MAs/CF. Moreover, the $^{13}$C NMR spectra of the electrolyzed products after 24 h MOR further indicate the superior HCOO$^-$ selectivity for Cu$_{SA}$-Rh MAs/CF and the generated CO$_3^{2-}$ as a by-product is almost negligible (Supplementary Fig. 16). It is worth mentioning that the Cu$_{SA}$-Rh MAs/CF possesses a superior MOR activity compared with Rhene-CF, Rh NPs-CF and CF (Supplementary Fig. 17a). Meanwhile, relative to the

standard potential (0.103 V vs RHE) of MOR[17], the overpotential of Cu$_{SA}$-Rh MAs/CF is 1.25 V (vs RHE) for reaching a current density of 20 mA cm$^{-2}$, which is lower than those of Rhene-CF (1.30 V vs RHE), Rh NPs-CF (1.31 V vs RHE) and CF (1.35 V vs RHE) (Supplementary Fig. 17b). Additionally, the MOR stability of Cu$_{SA}$-Rh MAs/CF was evaluated by the repeated *i-t* tests under a constant potential. Figure 4h shows that the Cu$_{SA}$-Rh MAs/CF consistently maintains the HCOO$^-$ FEs as high as ~90% during the five repeated cycles testing, indicating excellent stability for HCOO$^-$ production on Cu$_{SA}$-Rh MAs/CF. To investigate the charge transfer kinetics at the electrocatalytic interface, Electrochemical impedance spectroscopy (EIS) for various electrocatalysts was recorded in a 1 M KOH solution. As displayed in Supplementary Fig. 18, the Cu$_{SA}$-Rh MAs/CF (1.13 Ω) presents a smaller resistance ($R_{ct}$) value than Rhene-CF (1.96 Ω), Rh NPs-CF (3.34 Ω) and CF (9.68 Ω), revealing a rapid interfacial charge transfer on Cu$_{SA}$-Rh MAs/CF[52,53]. Besides, the electrochemical active surface areas (ECSAs) for various electrocatalysts were evaluated by the electrochemical double-layer capacitance ($C_{dl}$) calculated based on cyclic voltammetry (CV) curves (Supplementary Fig. 19). The $C_{dl}$ value for Cu$_{SA}$-Rh MAs/CF (49.9 mF cm$^{-2}$) was calculated to be higher than those of Rhene-CF (35.9 mF cm$^{-2}$), Rh NPs-CF (17.1 mF cm$^{-2}$) and

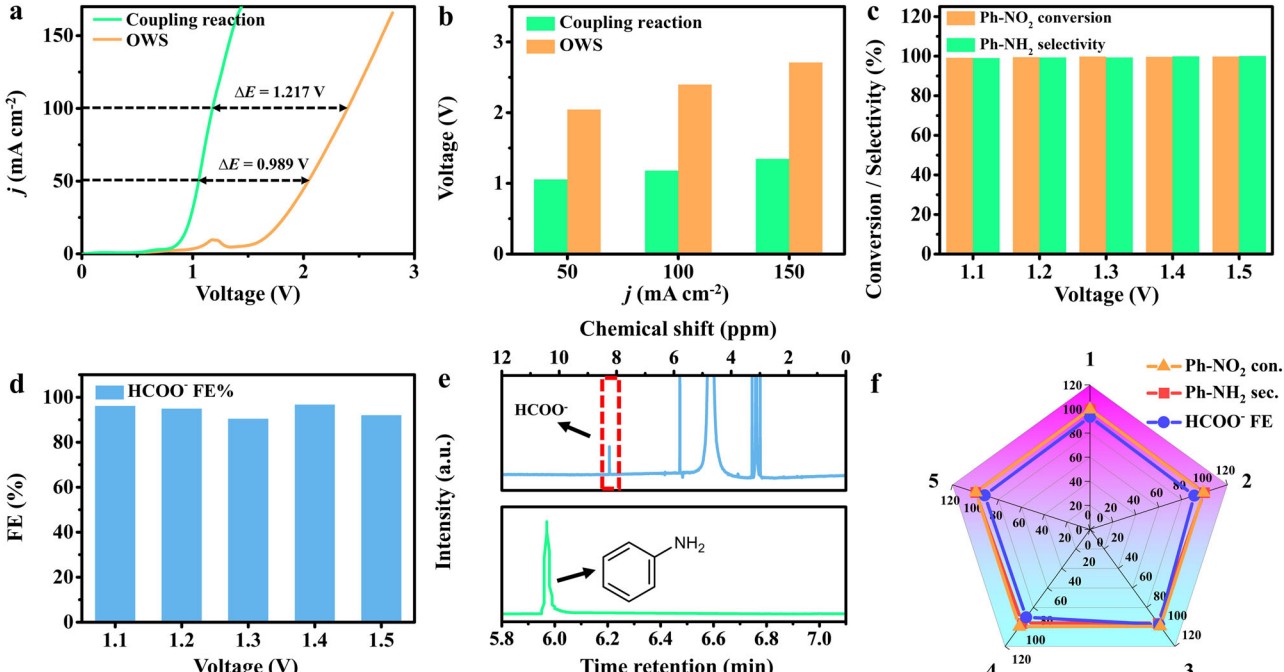

**Fig. 5 | Electrocatalytic performance of Ph-NO$_2$ ERR-MOR system. a** IR-corrected LSV curves and **b** corresponding voltages comparison for Cu$_{SA}$-Rh MAs/CF||Cu$_{SA}$-Rh MAs/CF (2.5 mg cm$^{-2}$) in the coupled Ph-NO$_2$ ERR-MOR system (cathode: 1 M KOH + 5 mM Ph-NO$_2$ solution, pH = 13.7 and anode: 1 M KOH + 4 M CH$_3$OH solution, pH = 13.7) and overall water splitting system (OWS, cathode and anode: 1 M KOH solutions, pH = 13.9). **c** Ph-NO$_2$ conversion and Ph-NH$_2$ selectivity for Cu$_{SA}$-Rh MAs/ CF||Cu$_{SA}$-Rh MAs/CF under various potentials at the cathode. **d** HCOO$^-$ FEs for Cu$_{SA}$-Rh MAs/CF||Cu$_{SA}$-Rh MAs/CF under various potentials at the anode. **e** GC result of products for cathodic Ph-NO$_2$ ERR and $^1$H NMR spectra of products for anodic MOR over Cu$_{SA}$-Rh MAs/CF||Cu$_{SA}$-Rh MAs/CF at 1.2 V. **f** Ph-NO$_2$ conversion, Ph-NH$_2$ selectivity and HCOO$^-$ FEs for Cu$_{SA}$-Rh MAs/CF||Cu$_{SA}$-Rh MAs/CF during the coupled reaction at 1.2 V measured for 5 successive cycles.

CF (2.1 mF cm$^{-2}$) (Supplementary Fig. 20), which reveals the rich active sites in Cu$_{SA}$-Rh MAs/CF due to the ultrathin metallene array structure and the introduction of isolated Cu single-atom.

### Electrocatalytic performance for Ph-NO$_2$ ERR-MOR system

Inspired by the above results of two half-reactions, a two-electrode system (Ph-NO$_2$ ERR-MOR) was constructed by employing the Cu$_{SA}$-Rh MAs/CF as cathode and anode. The constructed coupling system enables the simultaneous conversion of organic small molecules to targeted high-value-added chemicals and greatly optimizes energy efficiency. As shown in Fig. 5a, LSV curves of Cu$_{SA}$-Rh MAs/CF||Cu$_{SA}$-Rh MAs/CF were recorded in the Ph-NO$_2$ ERR-MOR system and overall water splitting (OWS) system. Apparently, the Ph-NO$_2$ ERR-MOR system is driven requiring a lower onset potential compared with OWS, indicating a faster reaction kinetics. The cell voltages (1.05, 1.18, and 1.35 V) of Ph-NO$_2$ ERR-MOR system are lower than OWS system (2.04, 2.40, and 2.71 V) at current densities of 50,100, and 150 mA cm$^{-2}$ (Fig. 5b), revealing the improved electrolysis efficiency and lower electrolysis energy consumption in the Ph-NO$_2$ ERR-MOR system. For evaluating the selectivity and activity of Cu$_{SA}$-Rh MAs/CF||Cu$_{SA}$-Rh MAs/CF in the Ph-NO$_2$ ERR-MOR system, the electrolyzed products of cathode and anode were collected and analyzed at various applied potentials. As presented in Fig. 5c, d, the cathodic Ph-NO$_2$ conversion and Ph-NH$_2$ selectivity are up to ~100% as well and the anodic HCOO$^-$ FEs also reaches over ~90% under various applied potentials, which indicates the efficient and selective targeted production of Ph-NH$_2$ and HCOO$^-$ at cathode and anode. Furthermore, almost no by-products are generated in the Ph-NO$_2$ ERR-MOR coupling reaction of Cu$_{SA}$-Rh MAs/ CF||Cu$_{SA}$-Rh MAs/CF (Fig. 5e). More encouragingly, the Ph-NO$_2$ conversion/Ph-NH$_2$ selectivity/HCOO$^-$ FE of Cu$_{SA}$-Rh MAs/CF||Cu$_{SA}$-Rh MAs/CF remain stable after 5 repeated cycles testing in the Ph-NO$_2$ ERR-MOR system (Fig. 5f). Furthermore, after stability testing, no significant degradation is observed for the morphology and structure of

Cu$_{SA}$-Rh MAs/CF (Supplementary Fig. 21a, b), and the crystal structure of Cu$_{SA}$-Rh MAs remains stable (Supplementary Fig. 21c, d). Notably, Supplementary Fig. 22 further reveals that the elemental composition and chemical state of Cu$_{SA}$-Rh MAs show no significant change after stability testing. These conclusions indicate superior stability for the Ph-NO$_2$ ERR-MOR coupling system constructed by the Cu$_{SA}$-Rh MAs/CF.

### DFT calculations

To further reveal the electronic structure effect between the isolated Cu single-atom and Rh host as well as the mechanism of enhanced electrocatalytic activity on Cu$_{SA}$-Rh MAs/CF, DFT calculations were carried out. Supplementary Fig. 23 shows an optimized geometric structure model for Cu$_{SA}$-Rh MAs. Based on the Bader charge calculation analysis (Supplementary Fig. 24 and Supplementary Table 2), the total net charges of Rh and Cu are 1.94 e and −1.94 e, respectively, which reveals that the electron transfer is from the Cu single-atom to the Rh host. Fig. 6a, b and Supplementary Fig. 25 reveal the accumulation of negative charge around the Rh host while Cu single-atom possesses positive charge property due to the loss of charge, which further indicate the electron-rich nature of Rh localization resulting from the introduction of isolated Cu single-atom. The projected partial density of states (PDOSs) of Cu$_{SA}$-Rh MAs show the strong overlapping of the Rh-4$d$ orbital with the Cu-3$d$ orbital (Fig. 6c), revealing the effective site-to-site electron transfer between the isolated Cu single-atom and the Rh host, which contributes to the optimization of electronic structures thus facilitating electrocatalytic reactions[54,55]. Moreover, as displayed in Fig. 6d, the $d$-band center of Rh-4$d$ orbitals for Cu$_{SA}$-Rh (111) (−1.79 eV) exhibits a slight negative shift compared with Rh (111) (−1.77 eV). It is notable that the PDOS for Rh-4$d$ of Cu$_{SA}$-Rh bulk (−1.82 eV) and Rh bulk (−1.49 eV) also reflect a similar trend (Supplementary Fig. 26). As shown in Supplementary Fig. 27 and

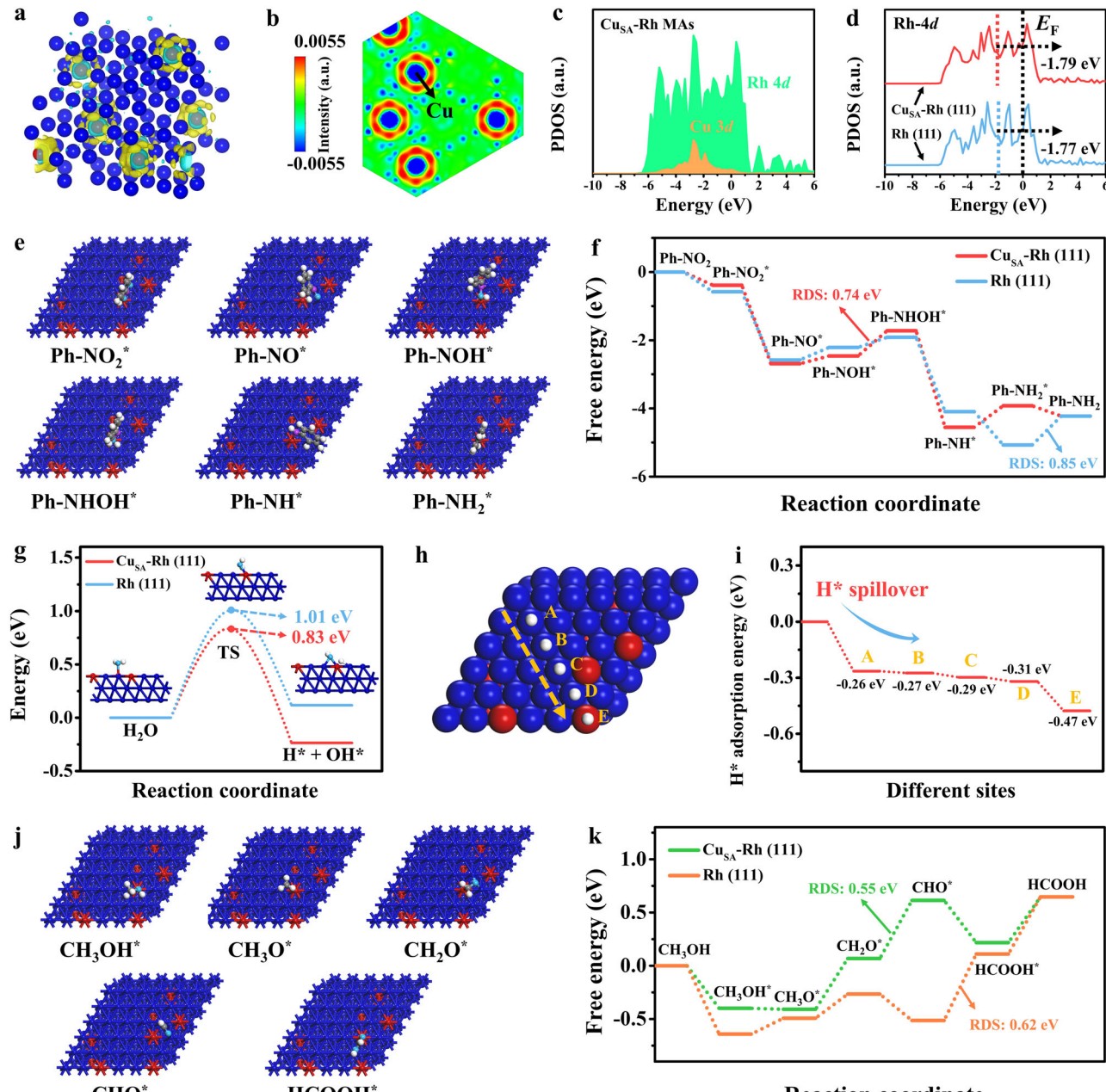

**Fig. 6 | DFT calculations analysis. a** Charge density difference on Cu$_{SA}$-Rh model. The blue and red spheres represent Rh and Cu atoms respectively, as well as the yellow and cyan indicate the charge depletion and accumulation areas. **b** Slice of charge density difference for Cu$_{SA}$-Rh bulk structure. **c** The PDOSs of Cu$_{SA}$-Rh. **d** The PDOSs for Rh-4$d$ of Rh (111) and Cu$_{SA}$-Rh (111) surfaces. **e** Optimized structures of Ph-NO$_2$ ERR intermediates on Cu$_{SA}$-Rh (111). The blue, red, gray, cyan, purple, and white spheres represent Rh, Cu, C, O, N, and H atoms respectively. **f** Comparison of free energy profiles for Ph-NO$_2$ ERR pathway on Cu$_{SA}$-Rh (111) and

Rh (111). **g** Calculated energy profiles of H$_2$O dissociation on Cu$_{SA}$-Rh (111) and Rh (111). **h** Optimized structures of adsorbed H* and **i** calculated H* adsorption energies variations at different sites on Cu$_{SA}$-Rh (111). The blue, red, and white spheres represent Rh, Cu, and H atoms respectively. **j** Optimized structures of MOR intermediates on Cu$_{SA}$-Rh (111). The blue, red, gray, cyan, and white spheres represent Rh, Cu, C, O, and H atoms respectively. **k** Comparison of free energy profiles for MOR pathway on Cu$_{SA}$-Rh (111) and Rh (111).

Supplementary Table 3, The electron transfer between Cu$_{SA}$-Rh and Ph-NO$_2$* (0.19 e) is smaller than that between Rh and Ph-NO$_2$* (0.25 e) owing to the electron interaction between Cu single-atom and the Rh host, which eventually can induce the decrease of electron interaction between Cu$_{SA}$-Rh and Ph-NO$_2$*, thus weakening the binding of Ph-NO$_2$* on Cu$_{SA}$-Rh[43]. Notably, there is electron accumulation in the antibonding orbitals for both Cu$_{SA}$-Rh and Rh, which favors the weakening of the bond energy for the N−O bond[43,56]. These conclusions indicate that the electron interaction between the isolated Cu single-atom and the Rh host causes a downshift of

the $d$-band center and a decrease in the electron interaction between the catalyst and adsorbate, thus promoting a fast conversion of the reactants to key intermediates as well as optimizing the desorption of the target product[8,43]. Moreover, Supplementary Fig. 28a and Supplementary Table 4 present the optimized adsorption structure and adsorption energy ($\Delta E$ads) of Ph-NO$_2$ on Rh (111) and Cu$_{SA}$-Rh (111) surfaces. The calculated Ph-NO$_2$ $\Delta E$ads of Cu$_{SA}$-Rh (111) is −1.08 eV, which is lower than that of Rh (111) (−1.27 eV). The weakened $\Delta E$ads of Ph-NO$_2$ on Cu$_{SA}$-Rh is beneficial for the rapid conversion of Ph-NO$_2$ to Ph-NOOH* by coupling with

the protons in $H_2O$, and the unstable Ph-NOOH* rapidly removes OH* to further form the stable and important intermediate Ph-NO*, thus enhancing the rapid protonation in the Ph-NO$_2$ ERR process[8]. Notably, Supplementary Fig. 28b and Supplementary Table 4 show that the $\Delta E$ads of the key intermediate Ph-NO* on Cu$_{SA}$-Rh (111) (−1.34 eV) is higher than that of Rh (111) (−1.28 eV), indicating that the intermediate Ph-NO* remains stable on Cu$_{SA}$-Rh and can be steadily converted to Ph-NOH* thus reducing the desorption of Ph-NO* to avoid the generation of undesired by-product azobenzene. Moreover, the free energy profiles of optimized intermediates for Ph-NO$_2$ ERR pathway reveal Ph-NOH*→Ph-NHOH* and Ph-NH$_2$*→Ph-NH$_2$ as the rate-determining step (RDS) for Cu$_{SA}$-Rh and Rh, respectively (Fig. 6e, f and Supplementary Fig. 29). Obviously, the Cu$_{SA}$-Rh exhibits a lower energy barrier (0.74 eV) on the RDS compared with Rh (0.85 eV) (Fig. 6f and Supplementary Table 5), further indicating the higher ability on Cu$_{SA}$-Rh for driving the Ph-NO$_2$ ERR to Ph-NH$_2$. To further investigate the enhanced mechanism of activity during the Ph-NO$_2$ hydrogenation with $H_2O$ as the hydrogen source on Cu$_{SA}$-Rh MAs, a series of DFT calculations were performed and analyzed. Figure 6g and Supplementary Fig. 30 display the calculated energy profiles of $H_2O$ dissociation process and corresponding optimized structures of the initial, transition, and final states. The energy barrier for $H_2O$ dissociation of Cu$_{SA}$-Rh (0.83 eV) is lower than that of Rh (1.01 eV) (Fig. 6g and Supplementary Table 6), indicating that the introduction of the Cu single-atom is beneficial for the dissociation of $H_2O$ on Cu$_{SA}$-Rh to facilitate the formation H* for Ph-NO$_2$ hydrogenation. Furthermore, the H* $\Delta E$ads of various sites (A to E) were further calculated for investigating the hydrogenation mechanism of H* with $H_2O$ as the hydrogen source on Cu$_{SA}$-Rh MAs (Fig. 6h, i). Visually, the variation of H* $\Delta E$ads from A to E sites on Cu$_{SA}$-Rh reveals a gradually increased H* adsorption (Fig. 6i and Supplementary Table 7), which indicates a H*-spillover process between the isolated Cu single-atom and the Rh host. This can be ascribed to local adsorption differences induced by electronic effect of Cu single-atom and Rh host. In the H*-spillover process (Supplementary Fig. 31), the formed H* from $H_2O$ dissociation on Rh host can spontaneously and rapidly migrate on the isolated Cu single-atom thus stabilizing H* for the hydrogenation process of intermediates, which is beneficial for inhibiting competing reaction HER and avoiding the H* accumulation on the Rh host to occupy the active sites thus maximizing the utilization of H* to promote synergistic electrocatalytic effects on Cu single-atom and Rh host for Ph-NO$_2$ ERR. For further investigating the MOR mechanism on Cu$_{SA}$-Rh MAs, the free energy profiles of optimized intermediates for MOR pathway were analyzed by the DFT calculations (Fig. 6j, k and Supplementary Fig. 32). It can be observed that the Cu$_{SA}$-Rh possesses a lower energy barrier for RDS (CH$_2$O*→CHO*, 0.55 eV) compared to the RDS (CHO*→HCOOH*, 0.62 eV) of Rh (Fig. 6k and Supplementary Table 8), indicating a more favorable MOR process on Cu$_{SA}$-Rh. Additionally, the CH$_3$OH-to-HCOO$^-$ is regarded as a continuous dehydrogenation process of CH$_3$OH, and the adsorption of the key intermediate CHO* is critical for the HCOO$^-$ synthesis. As shown in Supplementary Fig. 33 and Supplementary Table 9, the $\Delta E$ads of CHO* for Cu$_{SA}$-Rh (111) (−5.09 eV) is lower than that for Rh (111) (−5.33 eV). Meanwhile, Fig. 6k further shows that the energy barrier for the desorption step (HCOOH*→HCOOH) of Cu$_{SA}$-Rh (0.43 eV) is lower than that of Rh (0.54 eV). These conclusions reveal that the introduction of Cu single-atom can effectively reduce the $\Delta E$ads of key intermediate CHO* leading to the easier desorption of the target product from the Cu$_{SA}$-Rh MAs surface, which is conducive to facilitating the HCOO$^-$ formation and avoiding the unnecessary oxidation process. Moreover, the MOR mechanism on Cu$_{SA}$-Rh was speculated as illustrated in Supplementary Fig. 34. Due to the electron effect between Cu and Rh, the electron transfer from the Cu single-atom to Rh host

can activate electron-rich Rh (reaction 1).

$$Cu + Rh \rightarrow Cu(+)/Rh(-) \tag{1}$$

Initially, The CH$_3$OH tends to adsorb on the electron-rich Rh (reaction 2), and then the CHO* is produced through a series of dehydrogenation reactions on Cu$_{SA}$-Rh (reactions 3 and 4).

$$Rh(-)/CH_3OH + OH^- \rightarrow Rh(-)/CH_2OH^* + e^- + H_2O(l) \tag{2}$$

$$Rh(-)/CH_2OH^* + OH^- \rightarrow Rh(-)/CH_2O^* + e^- + H_2O(l) \tag{3}$$

$$Rh(-)/CH_2O^* + OH^- \rightarrow Rh(-)/CHO^* + e^- + H_2O(l) \tag{4}$$

Finally, the HCOO$^-$ is produced by nucleophilic attack of OH$^-$ adsorbed around the positively charged Cu against the electrophilic carbocation in CHO* and rapidly desorbs owing to the weak adsorption energy of CHO* (reaction 5).

$$Rh(-)/CHO^* + Cu(+)/OH^- \rightarrow Rh(-)/Cu(+) + HCOO^- + e^- + H2O(l) \tag{5}$$

Based on the above conclusion analysis, the superior Ph-NO$_2$ ERR and MOR activity of Cu$_{SA}$-Rh MAs/CF originates from the following points: Firstly, the stable security wall-like structure formed by the ultrathin metallene arrays provides sufficient active sites and abundant interlayer channels[45]. Secondly, the inherent defect-rich structure and low-crystalline regions of Cu$_{SA}$-Rh MAs/CF can induce unsaturated coordination metallic bonds and optimize the local electron structure[37,56]. Thirdly, the synergistic catalysis effect and H*-spillover effect between Cu single-atom and Rh host can optimize the catalytic reaction process, facilitate the stable and rapid conversion of reactants to intermediates as well as accelerate the desorption of target products. Fourthly, the Cu single-atom as effective adsorption sites can modulate the competition for adsorbate adsorption on Rh sites thus promoting electrocatalytic reactions.

## Discussion

In summary, we have successfully synthesized the Cu$_{SA}$-Rh MAs/CF with the isolated Cu single-atom dispersed on Rh metallene that possesses excellent electrocatalytic activity towards Ph-NO$_2$ ERR and MOR. The Cu$_{SA}$-Rh MAs/CF exhibits the low cell voltages (1.05 and 1.18 V) to achieve the current densities of 50 and 100 mA cm$^{-2}$ in the Ph-NO$_2$ ERR-MOR coupled system, and the cathodic Ph-NO$_2$ conversion/Ph-NH$_2$ selectivity are up to -100% as well as the anodic HCOO$^-$ FEs also reaches over -90%. The constructed coupled organic electrocatalytic conversion system achieves simultaneous conversion of low-value organic compounds to high-value chemicals at both cathode and anode as well as improves energy efficiency. The electron effect between Cu single-atom and Rh host and the catalytic reaction mechanism were further revealed by DFT calculations, in which the synergistic catalytic effect and H*-spillover effect are triggered by the local electronic structure change between the Cu single-atom and Rh host, which optimizes the catalytic reaction process and facilitate the rapid production of key intermediates and the rapid desorption of target products. This work provides a novel strategy for the design of SAAs catalysts applied in the sustainable green synthesis of high-value chemicals.

## Methods
### Materials and chemicals
Cu foam (CF) was provided by Changsha Lyrun Material Co., Ltd. Rhodium (II) acetate dimer (C$_8$H$_{12}$O$_8$Rh$_2$, Rh: 43−46%), rhodium

chloride (RhCl$_3$, 98%), potassium hydroxide (KOH, AR, 85%), methanol (CH$_3$OH, AR, 99.5%), ethylene glycol (EG, AR, 99%), N,N-dimethylformamide (DMF, AR, 99.5%), diethylenetriamine (DETA, AR, 99%) and maleic acid (≥99%) were purchased from Aladdin Industrial Corporation (China). Ethyl acetate (99%), nitrobenzene (Ph-NO$_2$, AR, 99%) and aniline (Ph-NH$_2$, ≥99.5%) were ordered from Shanghai Macklin Biochemical Co., Ltd. (China). Ethanol (C$_2$H$_5$OH, 95%) and hydrochloric acid (HCl, 37%) were acquired from Sinopharm Chemical Reagent Co., Ltd. (China). High-purity Ar gas (99.99%) was obtained from Hangzhou Special Gases Co., Ltd. (China).

## Synthesis of Cu$_{SA}$-Rh MAs/CF

For a typical synthesis of Cu$_{SA}$-Rh MAs/CF, the $1 \times 2$ cm$^2$ of CF was soaked in a 3 M HCl solution for 20 min, followed by washing several times with ethanol/water, and then dried at 50 °C in a vacuum oven for further utilization. Typically, 10 mg of C$_8$H$_{12}$O$_8$Rh$_2$ and 1 g of KOH were ultrasonically dissolved in a mixture of DMF (6 mL) and EG (4 mL). Then, 5 mL of DETA was dropped into the above solution to form a homogeneous solution, the obtained solution was transferred into a 25 mL Teflon-lined autoclave and the treated CF was placed into the autoclave. Afterward, the autoclave was heated to 200 °C and kept for 8 h. After naturally cooling to room temperature, the product was extracted from the autoclave and washed several times with water/ethanol. Finally, the product was dried in a vacuum oven at 60 °C for further characterization and electrochemical measurements.

## Synthesis of Rhene-CF

For a typical synthesis of Rhene-CF, 1 g of KOH and 10 mg of C$_8$H$_{12}$O$_8$Rh$_2$ were dissolved in a 30 mL vial containing 6 mL of DMF and 4 mL of EG. Then, 5 mL of DETA was slowly added to the above solution to form a homogeneous solution. The obtained solution was transferred to a 25 mL of Teflon-lined autoclave, followed by heating to 200 °C and maintaining for 1 h. After that, the product was obtained by centrifuging and washing several times with water/ethanol and dried in a vacuum oven at 60 °C for further utilization. Finally, the Rhene-CF was prepared by covering the prepared Rhene on CF.

## Synthesis of Rh NPs-CF

For a typical synthesis of Rh NPs-CF, 10 mg of RhCl$_3$ was dissolved in 30 mL of EG to form a homogeneous solution. The above solution was transferred to a 50 mL Teflon-lined autoclave and heated at 200 °C for 2 h. Then, the product was collected by centrifuging and washing several times with water/ethanol and dried in a vacuum oven at 60 °C for further utilization. Finally, the Rh NPs-CF was prepared by covering the prepared Rh NPs on CF.

## Material characterizations

Scanning electron microscopy (SEM) images were collected on a Zeiss Gemini 500. TEM images and selected area electron diffraction (SAED) data were collected on an FEI Tecnai G2 F30 (300 kV). Aberration-corrected high angle annular dark field scanning transmission electron microscopy (AC-HAADF-STEM) images, EELS, and elemental mapping data were collected on a Thermo Scientific Themis Z. Atomic force microscopy (AFM) was conducted on a Bruker Multimode 8. XRD data were recorded with a PANalytical Empyrean powder diffractometer using Cu Kα radiation and XPS data were collected on a Thermo ESCALAB 250XI. Rh $K$-edge and Cu $K$-edge XANES and EXAFS data were acquired from a BL14W1 beamlines at the Shanghai Synchrotron Radiation Facility (Shanghai, China). The inductively coupled plasma optical emission spectroscopy (ICP-OES) data were collected from a PerkinElmer ICP 2100. Nuclear magnetic resonance (NMR) spectra were collected on a Bruker Avance NEO 600 and gas chromatograph (GC) was conducted on an Agilent 8890/7000D.

## Electrochemical experiments

Electrochemical measurements were conducted on a CHI760E electrochemical workstation (Shanghai Chenhua Instrument Corporation, China) via a two/three-electrode system, where contains the reference electrode (Hg/HgO electrode), counter electrode (Pt foil and carbon rod), and working electrode. As a blinder-free electrode, the Cu$_{SA}$-Rh MAs/CF can be directly used as a working electrode, where Rhene-CF, Rh NPs-CF and CF serve as the comparison. Regarding cathodic Ph-NO$_2$ electroreduction reaction (Ph-NO$_2$ ERR), all electrochemical tests were conducted in an H-type cell with the Nafion 117 membrane separation, where the cathode chamber contains working electrodes and Hg/HgO electrode, and the anode chamber contains Pt foil. The cathode and anode electrolytes were a 1 M KOH + 5 mM Ph-NO$_2$ solution and a 1 M KOH solution, respectively. Before the Ph-NO$_2$ ERR test, Ar gas (99.99%) was continuously passed into the cathode chamber to purify the electrolyte for 20 min. More importantly, the cathode chamber needs to be continuously flooded with Ar gas during the whole Ph-NO$_2$ ERR tests. Regarding anodic methanol oxidation reaction (MOR), electrochemical tests were conducted in a single-chamber cell via a three-electrode system containing a working electrode, reference electrode (Hg/HgO electrode), and counter electrode (carbon rod). Regarding Ph-NO$_2$ ERR-MOR two-electrode system, the Cu$_{SA}$-Rh MAs/CF served as the cathode and anode electrodes, respectively. The cathode and anode electrolytes were a 1 M KOH + 5 mM Ph-NO$_2$ solution and a 1 M KOH + 4 M CH$_3$OH solution, respectively. All linear sweep voltammetry (LSV) curves were recorded at a scan rate of 5 mV s$^{-1}$ with 95% iR-compensation, and the current density for Ph-NO$_2$ ERR and MOR was acquired by normalizing to the geometric area of the CF. The applied potentials are converted into reversible hydrogen electrode (RHE) scale according to the Eq. (6):

$$E(vs.RHE) = E^{\theta}(Hg/HgO) + E(vs.Hg/HgO) + 0.059 \times pH. \qquad (6)$$

EIS was carried out in the range of 0.1 Hz to 100 kHz.

## Product analysis

For Ph-NO$_2$ ERR, the electrolyte was collected and extracted with ethyl acetate after $i$-$t$ tests. The extracted products were determined by comparing the GC retention times and mass spectra. The Ph-NO$_2$ conversion and Ph-NH$_2$ selectivity were acquired via the GC results analysis and calculated based on the following Eqs. (7) and (8):

$$\text{Conversion} = \frac{\text{mol of the consumed Ph-NO}_2}{\text{mole of the added Ph-NO}_2} \times 100\% \qquad (7)$$

$$\text{Selectivity} = \frac{\text{mol of the as-formed Ph-NH}_2}{\text{mole of the added Ph-NO}_2} \times 100\% \qquad (8)$$

For MOR, the electrolyte after $i$-$t$ tests was collected and analyzed via the NMR spectroscopy, where maleic acid served as an internal standard. The HCOO$^-$ FEs were calculated based on the following Eq. (9):

$$FE(\text{HCOO}^-) = \frac{N(\text{Production}) \times Z \times F}{Q} \times 100\% \qquad (9)$$

Where $N$ is the molar amount for the formed HCOO$^-$, $Z$ is the number of electrons transferred for the formed HCOO$^-$ ($Z = 4$), $F$ is the Faraday constant (96485 C mol$^{-1}$), $Q$ is the total amount of charge consumed.

## Computational methods

Details of the calculation method can be found in the Supplementary method.

## Data availability

The authors declare that all data supporting the findings of this study are available within the article and its Supplementary Information. The source data generated in this study are available in the figshare repository (https://doi.org/10.6084/m9.figshare.23994690.v1).

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

## Acknowledgements
This work was financially supported by the National Natural Science Foundation of China (No. 21972126, 21978264, 21905250, and 22278369), Natural Science Foundation of Zhejiang Province (No. LQ22B030012 and LQ23B030010), and China Postdoctoral Science Foundation (2021M702889).

## Author contributions
H.W. and Q.M. conceived the idea and designed the experiments. X.M. and W.W. conducted the synthesis and characterization of samples and electrochemical experiments. H.W. and L.W. supervised the project. K.D. and L.W. analyzed the XANES and EXAFS results. Z.W. and H.Y. performed and analyzed DFT calculations. H.W. and Q.M. analyzed the results and wrote the paper. K.D. and Y.X. helped with the revision of the paper.

## Competing interests
The authors declare no competing interests.
