## [Peer Review File · Nature Communications]

Atomically dispersed Cu coordinated Rh metallene arrays for simultaneously electrochemical aniline synthesis and biomass upgradingREVIEWER COMMENTS

Reviewer #1 (Remarks to the Author):

In this manuscript, the authors report a simple and rapid one-step solvothermal synthesis of CuSA-Rh MAs/CF. As a bifunctional electrocatalyst, CuSA-Rh MAs/CF exhibits excellent electrocatalytic activity towards Ph-NO₂ ERR and MOR. For the constructed Ph-NO₂-ERR-MOR coupled electrocatalytic system, only a low voltage of 1.05/1.18 V can achieve a current density of 50/100 mA cm⁻² of Ph-NO₂-to-Ph-NH₂. For the efficient conversion of methanol to formic acid, the Ph-NO₂ conversion and Ph-NH₂ selectivity are as high as about 100%, and HCOO-FE outperforms CuSA-Rh MAs/CF by about 90%. In addition, density functional theory (DFT) calculations further reveal the synergistic catalytic effect and H* spillover effect induced by local electronic changes between isolated Cu single atoms and Rh hosts, facilitating the rapid conversion of reactants to key intermediates. And the target product is quickly desorbed, thereby improving the electrocatalytic reaction activity and the selectivity of the target product. The authors discuss in detail in the article; however, some problems should be improved. It can be acceptable for publication after major revisions.

(1) This article is not innovative enough, some similar work has already been done, such as Mao, Qiqi, et al. "Sulfur Vacancy-Rich Amorphous Rh Metallene Sulfide for Electrocatalytic Selective Synthesis of Aniline Coupled with Efficient Sulfion Degradation." ACS nano (2022).

(2) The authors need to further confirm the existence of copper-copper atoms by EELS with high-resolution transmission electron microscopy.

(3) Some recently published references could be cited to enrich the introduction part, such as 10.1016/j.cjcs.2023.100035 and 10.1002/adma.202302007.

(4) In the XRD pattern of CuSA-Rh MAs, the diffraction peaks of CuSA-Rh MAs are obviously shifted relative to the JCPDS card of Rh. What causes this? What is the relative atomic content of CuSA? The authors need to further give the content value through ICP and other analysis methods.

(5) There are grammatical and spelling errors in some parts of the article, which need further confirmation by the authors.

(6) The authors need to provide the XRD\XPS\HRTEM data of the CuSA-Rh MAs/CF sample after the catalytic performance test to prove the stability of the catalyst.

Reviewer #2 (Remarks to the Author):

By combining experiments and DFT calculations, the authors proposed the Cu dispersed Rh metallene arrays as new electrocatalysts for both Ph-NO₂ ERR and MOR. However, there are several important computational evidence missing in their work, making it hard to be accepted.

1. How did they consider and construct the computational model in Fig. S18? Is it consistent with their experimental observation?

2. For Fig. S19, one cannot tell which one is Rh or Cu.

3. The coloring information is missing in Fig. S20, making it impossible to tell the charge transfer direction.

4. The quality of Fig. 6 should be further improved for clarity. H and I have repeated information, and they can be merged.

5. For the d-band center (they should mention that it is for the Rh atom to avoid misleading), a slight change from -1.77 to -1.79 eV cannot be considered as "a significant negative shift". It is not clear why a downshift of d-band center and occupancy of anti-bonding orbital "thus promoting a fast conversion of the reactants to key intermediates".

6. The most important one is, besides the adsorption energy calculations, the whole free energy profiles (including water dissociation) should be provided to support their proposed mechanisms of the two reactions. Detailed comparison should be done between those with and without the Cu. All the related data leading to these energy profiles should be summarized in supporting information.

Reviewer #3 (Remarks to the Author):

In this paper, the authors synthesized Cu single-atom dispersed Rh metallene arrays on Cu foam and use them as electrocatalysts for the reduction of nitrobenzene and oxidation of methanol. Characterization of the catalyst was carried out using a combination of several techniques. The electrochemical reaction yielded the target products (aniline and formate) with high selectivity. A reaction mechanism is proposed using DFT calculations.

The reviewer carefully considered the paper. The publication in Nature Communications would require greater scholarly significance than in previous studies. However, the synthesis of aniline by electro-oxidation of nitrobenzene and the synthesis of formic acid by oxidation of methanol have already been reported; thus, there is no novelty in the reactions. Although there may not be any studies combining these two reactions, as an electrolysis process, it is merely a combination of two known reactions.

The remaining novelty may lie in the performance of the catalyst. However, there are still doubts regarding the significance of the catalytic activity. While Table S2 compares the results with previous studies, it is important to note that the electrochemical reactions being compared are fundamentally different. Therefore, comparing voltages among different reactions does not hold significant meaning. Additionally, it should be noted that the majority of catalysts in previous studies were transition metal catalysts, while this study utilizes noble metals (which generally exhibit higher catalytic activity). Hence, making such comparisons may not be meaningful in this context.

For the above reasons, the reviewer has determined that this paper is not acceptable for publication in Nature Communications.
Other comments are noted below.

The effect of Cu single-atom also seems unclear; the effect of Cu single-atom cannot be discussed without preparing Rh metallene arrays without Cu single-atom and comparing their catalytic activity. The comparison with Rh nanoparticles is made in the paper, but as mentioned in the paper, the surface areas are different between Rh NPs and Rh metallene. Therefore, the comparison may not be appropriate for discussing the effect of Cu single atoms due to the differences in surface area.

The authors stated the electronic interaction between Cu and Rh, but what is the density of Cu in the Rh metallene? If Cu is sparsely present on Rh metallene, there would be Rh atoms that do not interact with Cu. What is the percentage of Rh that the effect of Cu does reach?

There is a lack of discussion from an electrochemical point of view. At the very least, the standard redox potential should be given and overvoltages should be discussed.

Finally, the authors would need to check references. In the 52 references, only 4 papers are published by different nationalities than the authors. If appropriate works related to this study are cited, the reference list is no problem, but the reviewer is concerned about the large nationality bias in the reference list.

Point-to-Point Response to Reviewers

Reviewer #1

Comment: In this manuscript, the authors report a simple and rapid one-step solvothermal synthesis of Cu_{SA}-Rh MAs/CF. As a bifunctional electrocatalyst, Cu_{SA}-Rh MAs/CF exhibits excellent electrocatalytic activity towards Ph-NO₂ ERR and MOR. For the constructed Ph-NO₂-ERR-MOR coupled electrocatalytic system, only a low voltage of 1.05/1.18 V can achieve a current density of 50/100 mA cm⁻² of Ph-NO₂-to-Ph-NH₂. For the efficient conversion of methanol to formic acid, the Ph-NO₂ conversion and Ph-NH₂ selectivity are as high as about 100%, and HCOO⁻ FE outperforms Cu_{SA}-Rh MAs/CF by about 90%. In addition, density functional theory (DFT) calculations further reveal the synergistic catalytic effect and H* spillover effect induced by local electronic changes between isolated Cu single atoms and Rh hosts, facilitating the rapid conversion of reactants to key intermediates. And the target product is quickly desorbed, thereby improving the electrocatalytic reaction activity and the selectivity of the target product. The authors discuss in detail in the article; however, some problems should be improved. It can be acceptable for publication after major revisions.

Reply: We thank Referee #1 for his/her valuable time reviewing our manuscript. We also appreciate his/her positive comments.

Comment 1-1: This article is not innovative enough, some similar work has already been done, such as Mao, Qiqi, et al. "Sulfur Vacancy-Rich Amorphous Rh Metallene Sulfide for Electrocatalytic Selective Synthesis of Aniline Coupled with Efficient Sulfion Degradation." ACS nano (2022).

Reply 1-1: Although Ph-NO₂ ERR have been reported so far, in this work, we constructed for the first time Cu single-atom coordinated Rh metallene array structure and proposed the unique mechanisms (H*-spillover effect and synergistic catalytic effect) to enhance the activity and selectivity of Ph-NO₂ ERR and MOR, which is different from the previously reported work. More importantly, we constructed the Cu_{SA}-Rh MAs/CF as a bifunctional catalyst applied in a novel Ph-NO₂ ERR-MOR two-electrode system, which achieves the simultaneous generation of high-value chemicals at the cathode and anode. Meanwhile, we proved the unique mechanism of the introduction of Cu single-atom on the enhancement of catalytic activity by a series of experiments and DFT calculations. The design of this single-atom alloy metallene structure and the use of H*-

spillover effect and synergistic catalytic effect are promising strategies for applying in other organic electrocatalytic conversion reactions. Thus, this work is innovative. Please see Page 19.

Revision made in manuscript: Please see Page 19. We have added the following revisions to the manuscript and highlighted them in the marked-up revised manuscript.

The superior Ph-NO₂ ERR and MOR activity of Cu_{SA}-Rh MAs/CF originates from the following points: Firstly, the stable security wall-like structure formed by the ultrathin metallene arrays provides sufficient active sites and abundant interlayer channels.⁴⁵ Secondly, the inherent defect-rich structure and low-crystalline regions of Cu_{SA}-Rh MAs/CF can induce unsaturated coordination metallic bonds and optimize the local electron structure.^{37,46} Thirdly, the synergistic catalysis effect and H*-spillover effect between Cu single-atom and Rh host can optimize the catalytic reaction process, facilitate the stable and rapid conversion of reactants to intermediates as well as accelerate the desorption of target products. Fourthly, the Cu single-atom as effective adsorption sites can modulate the competition for adsorbate adsorption on Rh sites thus promoting electrocatalytic reactions.

Comment 1-2: The authors need to further confirm the existence of copper-copper atoms by EELS with high-resolution transmission electron microscopy.

Reply 1-2: Thanks for the value suggestion. We have confirmed the existence of Rh and Cu elements by EELS analysis. As revealed by the EELS of Cu_{SA}-Rh MAs, the energy loss peak around 498.8 eV can be assigned to the Rh M electron transition (Please see Ref. 46. Adv. Mater. 2020, 32, 1908521 and Ref. 47. Adv. Mater. 2021, 33, 2007894), and the energy loss peak around 933.1 eV can be assigned to the Cu L electron transition (Please see Ref. 48. Appl. Surf. Sci. 2021, 563, 150318). The EELS data of Cu_{SA}-Rh MAs further reveal the existence of Rh and Cu elements. Please see Page 7 and Ref. 46, 47, 48 and Fig. S6.

Revision made in manuscript: Please see Pages 7 and Ref. 46, 47, 48. We have added the following revisions to the manuscript and highlighted them in the marked-up revised manuscript.

Furthermore, as revealed by the electron energy loss spectrum (EELS) of Cu_{SA}-Rh MAs (**Fig. S6**), the energy loss peak around 498.8 eV can be assigned to the Rh M electron transition (**Fig. S6a**),^{46,47} and the energy loss peak around 933.1 eV in **Fig. S6b** can be assigned to the Cu L electron transition.⁴⁸ The EELS data of Cu_{SA}-Rh MAs further reveal the existence of Rh and Cu elements.

46. Li Z, *et al.* Stable rhodium (IV) oxide for alkaline hydrogen evolution reaction. *Adv. Mater.* **32**, 1908521 (2020).
47. Zhong W, *et al.* RhSe₂: A superior 3D electrocatalyst with multiple active facets for hydrogen evolution reaction in both acid and alkaline solutions. *Adv. Mater.* **33**, 2007894 (2021).
48. Morales MR, Lajaunie L, Calvino JJ, Cauqui MÁ, Cadus LE, Hernández-Garrido JC. In-depth structural and analytical study of the washcoating layer of a Mn-Cu monolithic catalyst using STEM-FIB, EDX and EELS. Insights into stability under working conditions. *Appl. Surf. Sci.* **563**, 150318 (2021).

Revision made in supporting information: Please see Page S8 and Fig. S6. We have added the following revisions to the supporting information.

Fig. S6. (a and b) The EELS spectra of Rh and Cu in Cu_{SA}-Rh MAs.

Comment 1-3: Some recently published references could be cited to enrich the introduction part, such as 10.1016/j.cjsc.2023.100035 and 10.1002/adma.202302007.

Reply 1-3: Thanks for the value suggestion. We have cited the mentioned important references in the manuscript. Please see Pages 13 and Ref. 52, 53

Revision made in manuscript: Please see Page 13 and Ref. 52, 53. We have added the following revisions to the manuscript and highlighted them in the marked-up revised manuscript.

The Cu_{SA}-Rh MAs/CF (1.13 Ω) presents a smaller resistance (R_{ct}) value than Rhene-CF (1.96 Ω), Rh NPs-CF (3.34 Ω) and CF (9.68 Ω), revealing a rapid interfacial charge transfer on Cu_{SA}-Rh MAs/CF.^{52,53}

52. Wang Z, *et al.* Nontrivial topological surface states in Ru₃Sn₇ toward wide pH-range hydrogen evolution reaction. *Adv. Mater.*, 2302007 (2023).

53. Wang X, Zhang J, Wang Z, Lin Z, Shen S, Zhong W. Fabricating Ru single atoms and clusters

on CoP for boosted hydrogen evolution reaction. *Chinese J. Struc. Chem.* **42**, 100035 (2023).

Comment 1-4: In the XRD pattern of Cu_{SA}-Rh MAs, the diffraction peaks of Cu_{SA}-Rh MAs are obviously shifted relative to the JCPDS card of Rh. What causes this? What is the relative atomic content of Cu_{SA}? The authors need to further give the content value through ICP and other analysis methods.

Reply 1-4: Thanks for the valuable question. We have provided a detailed explanation about the negative shift of the diffraction peaks of Cu_{SA}-Rh MAs. The characteristic peaks of Cu_{SA}-Rh MAs exhibit a negative shift compared with the Rh JCPDS card, originating from the curved 2D geometrical structure and the introduction of Cu atoms. A similar phenomenon was also reported in other metallene catalysts (Ref. 38. *Nature* 2019, 574, 81-85). Moreover, we have analyzed the content of Cu single-atom in Cu_{SA}-Rh MAs by TEM-EDS and ICP-OES tests. The Rh/Cu atomic ratio was further determined to be approximately 93.6/6.4 via the TEM-EDS, the mass ratio (Rh/Cu = 95.1/4.9) and atomic ratio (Rh/Cu = 92.6/7.4) of Rh/Cu in Cu_{SA}-Rh MAs were further analyzed by ICP-OES, which is close to the results obtained from TEM-EDS. Please see Pages 6, 7 and Ref. 38 and Figs. S4, S5.

Revision made in manuscript: Please see Pages 6, 7 and Ref. 38. We have added the following revisions to the manuscript and highlighted them in the marked-up revised manuscript.

The Rh/Cu atomic ratio was further determined to be approximately 93.6/6.4 via the TEM energy dispersive X-ray spectroscopy (TEM-EDS, **Fig. S4**). The mass ratio (Rh/Cu = 95.1/4.9) and atomic ratio (Rh/Cu = 92.6/7.4) of Rh/Cu in Cu_{SA}-Rh MAs were further analyzed by inductively coupled plasma optical emission spectroscopy (ICP-OES) (**Fig. S5**), which is close to the results obtained from TEM-EDS.

In the X-ray diffraction (XRD) pattern (**Fig. 1h**), the characteristic peaks of Cu_{SA}-Rh MAs can be assigned to a typical face-centered cube (*fcc*) metallic Rh phase (No. 05-0685). Notably, the characteristic peaks of Cu_{SA}-Rh MAs exhibit a negative shift compared with the Rh JCPDS card, originating from the curved 2D geometrical structure and the introduction of Cu atoms.³⁸

38. Luo M, *et al.* PdMo bimetallene for oxygen reduction catalysis. *Nature* **574**, 81-85 (2019).

Revision made in supporting information: Please see Page S7 and Figs. S4, S5. We have added the following revisions to the supporting information.

Fig. S4. EDX spectrum of Cu_{SA}-Rh MAs.

Fig. S5. (a) Mass ratio and (b) atom ratio of Rh and Cu elements in Cu_{SA}-Rh MAs obtained from ICP-OES.

Comment 1-5: There are grammatical and spelling errors in some parts of the article, which need further confirmation by the authors.

Reply 1-5: Thanks for the value suggestion. We have carefully checked the whole manuscript and corrected the grammatical and spelling errors in the manuscript.

Comment 1-6: The authors need to provide the XRD\XPS\HRTEM data of the Cu_{SA}-Rh MAs/CF sample after the catalytic performance test to prove the stability of the catalyst.

Reply 1-6: Thanks for the value suggestion. We have provided the XRD\XPS\HRTEM data of the Cu_{SA}-Rh MAs/CF to prove the stability of the catalyst. Based on the results of SEM, TEM and HRTEM images, after stability testing, no significant degradation is observed for the morphology and structure of Cu_{SA}-Rh MAs/CF, and the crystal structure of Cu_{SA}-Rh MAs remains stable. Moreover, XPS results before and after stability testing illustrate that the elements composition and chemical state for Cu_{SA}-Rh MAs remain stable after stability testing. Please see Page 15 and Figs. S21-S22.

Revision made in manuscript: Please see Page 15. We have added the following revisions to the manuscript and highlighted them in the marked-up revised manuscript.

Furthermore, after stability testing, no significant degradation is observed for the morphology and structure of $\text{Cu}_{\text{SA}}\text{-Rh MAS/CF}$ (Figs. S21a-S21b), and the crystal structure of $\text{Cu}_{\text{SA}}\text{-Rh MAS}$ remains stable (Figs. S21c-S21d). Notably, Fig. S22 further reveals that the elemental composition and chemical state of $\text{Cu}_{\text{SA}}\text{-Rh MAS}$ show no significant change after stability testing. These conclusions indicate a superior stability for the Ph-NO_2 ERR-MOR coupling system constructed by the $\text{Cu}_{\text{SA}}\text{-Rh MAS/CF}$.

Revision made in supporting information: Please see Pages S14-S15 and Figs. S21-S22. We have added the following revisions to the supporting information.

Fig. S21. (a) SEM image of $\text{Cu}_{\text{SA}}\text{-Rh MAS/CF}$ after stability testing. (b) TEM, and (c) HRTEM images of $\text{Cu}_{\text{SA}}\text{-Rh MAS}$ after stability testing. (d) XRD patterns of $\text{Cu}_{\text{SA}}\text{-Rh MAS}$ before and after stability testing on cathode and anode.

Fig. S22. XPS spectra of (a) Rh 3d and (b) Cu 2p for $\text{Cu}_{\text{SA}}\text{-Rh MAS}$ before and after stability testing on cathode and anode.

Reviewer #2

Comment: By combining experiments and DFT calculations, the authors proposed the Cu dispersed Rh metallene arrays as new electrocatalysts for both Ph-NO₂ ERR and MOR. However, there are several important computational evidence missing in their work, making it hard to be accepted.

Reply: We thank Referee #2 for his/her valuable time reviewing our manuscript. We also appreciate his/her comments. We have supplemented the important DFT calculations for this work.

Comment 2-1: How did they consider and construct the computational model in Fig. S18? Is it consistent with their experimental observation?

Reply 2-1: Thanks for the valuable question. We have provided the method for constructing the computational model. Firstly, based on the experimental results analysis of AC-HAADF-STEM image, 3D topographic atom image, XANES and EXAFS tests, the Cu is present as a single-atom on Rh host and is mainly coordinated by the Cu-Rh bonds. Secondly, the XRD result indicates that the characteristic peaks of Cu_{SA}-Rh MAs can be assigned to a typical *fcc* metallic Rh phase (No. 05-0685), proving the formation of an alloy structure. These experimental results indicate the single-phase *fcc* crystal structure and single-atom alloy structure for Cu_{SA}-Rh MAs. Based on the above experimental results analysis, in the DFT simulation process, the Rh model is firstly constructed, and the Cu single-atom is induced with a random distribution, we used the *fcc* cubic phase of the original Rh as a template to build a 3×3×3 supercell containing 108 atoms for Cu_{SA}-Rh MAs, the atom ratio of Rh : Cu is approximately 93 : 7. Therefore, the constructed model is consistent with the experimental results. Please see Pages 7, 8, 9 and Page S5 and Fig. 1h and Figs 2a-2d and Figs 2h, 2i, 2k, 2l.

Revision made in manuscript: Please see Pages 7, 8, 9 and Fig. 1h and Figs 2a-2d and Figs 2h, 2i, 2k, 2l. We have added the following revisions to the manuscript and highlighted them in the marked-up revised manuscript.

In the X-ray diffraction (XRD) pattern (**Fig. 1h**), the characteristic peaks of Cu_{SA}-Rh MAs can be assigned to a typical face-centered cube (*fcc*) metallic Rh phase (No. 05-0685).

The Cu single-atom was analyzed by AC-HAADF-STEM image and 3D topographic atom images. **Fig. 2a** shows that some individual dark dots (Cu atoms) can be observed on the surface of Cu_{SA}-Rh MAs owing to the lower atomic number of Cu (29) compared with Rh (45). The low-intensity dots

also indicate the dispersion situation of the isolated Cu atoms on the surface of $\text{Cu}_{\text{SA}}\text{-Rh}$ MAs (Figs. 2b, c). The corresponding integrated pixel intensity profile also illustrates the isolated low-intensity Cu atoms dispersed surrounding the high-intensity Rh atoms on the crystal surface (Fig. 2d), further proving the presence of isolated Cu single-atom on the $\text{Cu}_{\text{SA}}\text{-Rh}$ MAs.

Notably, based on the WT spectra analysis for $\text{Cu}_{\text{SA}}\text{-Rh}$ MAs and Rh foil (Fig. 2i), the Rh-Rh/Cu-Rh intensity maximum of $\text{Cu}_{\text{SA}}\text{-Rh}$ MAs exhibits a negative shift of about 0.33 \AA^{-1} compared with Rh-Rh intensity maximum of Rh foil, which is induced by the coordination of the Cu-Rh bond.

The Cu *K*-edge EXAFS spectra show the distinct peak of $\text{Cu}_{\text{SA}}\text{-Rh}$ MAs at 2.43 \AA ascribed to the Cu-Rh bond, obviously distinct with Cu-Cu (2.23 \AA) and Cu-O (1.51 \AA) bands (Fig. 2k), revealing the presence of dispersed Cu single-atom.

As displayed in Fig. 2l, The WT spectra for $\text{Cu}_{\text{SA}}\text{-Rh}$ MAs, Rh foil and Rh_2O_3 further indicate the presence of isolated Cu single-atom, where the intensity maximum ($\sim 10.31 \text{ \AA}^{-1}$) of $\text{Cu}_{\text{SA}}\text{-Rh}$ MAs is ascribed to the Cu-Rh bond compared with Cu-Cu ($\sim 8.01 \text{ \AA}^{-1}$) and Cu-O ($\sim 6.76 \text{ \AA}^{-1}$) intensity maximums.

Fig. 1. (a) SEM image of $\text{Cu}_{\text{SA}}\text{-Rh}$ MAs/CF. (b) TEM image, (c) 3D view for the corresponding AFM image, and (d) AC HAADF-STEM image and the corresponding elemental mapping images of $\text{Cu}_{\text{SA}}\text{-Rh}$ MAs. (e) AC-HAADF-STEM image of $\text{Cu}_{\text{SA}}\text{-Rh}$ MAs. (f) The integrated pixel intensity profile for the selected red region in (e). (g) AC-HAADF-STEM image of $\text{Cu}_{\text{SA}}\text{-Rh}$ MAs. (h) XRD pattern of $\text{Cu}_{\text{SA}}\text{-Rh}$ MAs and the inset in (h) displays the corresponding SAED pattern of $\text{Cu}_{\text{SA}}\text{-Rh}$ MAs.

Fig. 2. (a) AC-HAADF-STEM image of $\text{Cu}_{\text{S}_A}\text{-Rh}$ MAs and the inset in (a) displays the magnified image of the selected white region in (a). (b, c) 3D topographic atom images of $\text{Cu}_{\text{S}_A}\text{-Rh}$ MAs. (d) The integrated pixel intensity profile for the selected region in (b). (e) Rh 3d XPS and (f) Cu 2p XPS spectra of $\text{Cu}_{\text{S}_A}\text{-Rh}$ MAs. (g) Rh K -edge XANES spectra, (h) Fourier transformed EXAFS spectra and (i) EXAFS wavelet transform diagrams of Rh foil, Rh_2O_3 and $\text{Cu}_{\text{S}_A}\text{-Rh}$ MAs. (j) Cu K -edge XANES spectra, (k) Fourier transformed EXAFS spectra and (l) EXAFS wavelet transform diagrams of Cu foil, CuO and $\text{Cu}_{\text{S}_A}\text{-Rh}$ MAs.

Revision made in supporting information: Please see Page S5. We have added the following revisions to the supporting information.

Based on experimental results, the $\text{Cu}_{\text{S}_A}\text{-Rh}$ MAs display the single-phase *fcc* crystal structure and single-atom alloy structure. In the DFT simulation process, the Rh model is firstly constructed, and the Cu single-atom is induced with a random distribution, we used the *fcc* cubic phase of the original Rh as a template to build a $3\times 3\times 3$ supercell containing 108 atoms for $\text{Cu}_{\text{S}_A}\text{-Rh}$ MAs, the atom ratio of Rh : Cu is approximately 93 : 7.

Comment 2-2: For Fig. S19, one cannot tell which one is Rh or Cu.

Reply 2-2: Thanks for the value suggestion. We have provided a detailed description for Fig. S24 (previously Fig. S19). Fig. S24a shows the net charge obtained from Bader charge analysis marked on Rh (blue balls) and Cu atoms (red balls). Fig. S24b shows the Bader charge of $\text{Cu}_{\text{S}_A}\text{-Rh}$, the balls

marked by red circles represent the tendency for electron transfer on Cu single-atom and the unmarked balls represent the tendency for electron transfer on Rh host. The analysis of electron transfer tendency on Cu and Rh atoms was obtained based on Fig. S24a. Please see Page S16 and Fig. S24.

Revision made in supporting information: Please see Page S16 and Fig. S24. We have added the following revisions to the supporting information.

Fig. S24. (a) Net charge obtained from Bader charge analysis marked on Rh (blue balls) and Cu atoms (red balls). (b) Bader charge of $\text{Cu}_{\text{SA}}\text{-Rh}$ (the balls marked by red circles represent the tendency for electron transfer on Cu single-atom and the unmarked balls represent the tendency for electron transfer on Rh host).

Comment 2-3: The coloring information is missing in Fig. S20, making it impossible to tell the charge transfer direction.

Reply 2-3: Thanks for the value suggestion. We have added the coloring information for Fig. S25 (previously Fig. S20) to show the charge transfer direction. Please see Page S16 and Fig. S25.

Revision made in supporting information: Please see Page S16 and Fig. S25. We have added the following revisions to the supporting information.

Fig. S25. (a) Main view, (b) side view and (c) top view of charge density difference on $\text{Cu}_{\text{SA}}\text{-Rh}$. The blue and red spheres represent Rh and Cu atoms respectively, as well as the yellow and cyan indicate the charge depletion and accumulation areas.

Comment 2-4: The quality of Fig. 6 should be further improved for clarity. H and I have repeated information, and they can be merged.

Reply 2-4: Thanks for the value suggestion. We have improved the resolution of Fig. 6 and merged the repeated information. Please see Page 33 and Fig. 6 and Fig. S33.

Revision made in manuscript: Please see Page 33 and Fig. 6. We have added the following revisions to the manuscript and highlighted them in the marked-up revised manuscript.

Fig. 6. (a) Charge density difference on $\text{Cu}_{\text{SA}}\text{-Rh}$ model. The blue and red spheres represent Rh and Cu atoms respectively, as well as the yellow and cyan indicate the charge depletion and accumulation areas. (b) Slice of charge density difference for $\text{Cu}_{\text{SA}}\text{-Rh}$ bulk structure. (c) The PDOSs of $\text{Cu}_{\text{SA}}\text{-Rh}$. (d) The PDOSs for Rh-4d of Rh (111) and $\text{Cu}_{\text{SA}}\text{-Rh}$ (111) surfaces. (e) Optimized structures of Ph- NO_2 ERR intermediates on $\text{Cu}_{\text{SA}}\text{-Rh}$ (111). The blue, red, gray, cyan, purple, and white spheres represent Rh, Cu, C, O, N, and H atoms respectively. (f) Comparison of free energy profiles for Ph- NO_2 ERR pathway on $\text{Cu}_{\text{SA}}\text{-Rh}$ (111) and Rh (111). (g) Calculated energy profiles of H_2O dissociation on $\text{Cu}_{\text{SA}}\text{-Rh}$ (111) and Rh (111). (h) Optimized structures of adsorbed H^* and (i) calculated H^* adsorption energies variations at different sites on $\text{Cu}_{\text{SA}}\text{-Rh}$ (111). The blue, red, and white spheres represent Rh, Cu, and H atoms respectively. (j) Optimized

structures of MOR intermediates on Cu_{SA}-Rh (111). The blue, red, gray, cyan, and white spheres represent Rh, Cu, C, O, and H atoms respectively. (k) Comparison of free energy profiles for MOR pathway on Cu_{SA}-Rh (111) and Rh (111).

Revision made in supporting information: Please see Page S19 and Fig. S33. We have added the following revisions to the supporting information.

Fig. S33. Calculated adsorption energies of CHO* on Rh (111) and Cu_{SA}-Rh (111).

Comment 2-5: For the d-band center (they should mention that it is for the Rh atom to avoid misleading), a slight change from -1.77 to -1.79 eV cannot be considered as “a significant negative shift”. It is not clear why a downshift of d-band center and occupancy of anti-bonding orbital “thus promoting a fast conversion of the reactants to key intermediates”.

Reply 2-5: Thanks for the value suggestion. We have improved and revised the description and explanation of *d*-band center as well as analyzed the effect of the downshift of the *d*-band center and the electron occupation of antibonding orbitals on the catalytic reaction in detail. Firstly, the *d*-band center of Rh-4d orbitals for Cu_{SA}-Rh (111) (-1.79 eV) exhibits a slight negative shift compared with Rh (111) (-1.77 eV) and it is notable that the PDOS for Rh-4d of Cu_{SA}-Rh bulk (-1.82 eV) and Rh bulk (-1.49 eV) also reflect the similar trend. The downshift of the *d*-band center facilitates the weakening of reactant adsorption. This was also confirmed by the calculation of the adsorption energy (ΔE_{ads}) of Ph-NO₂. The calculated Ph-NO₂ ΔE_{ads} of Cu_{SA}-Rh (111) is -1.08 eV, which is lower than that of Rh (111) (-1.27 eV). The weakened ΔE_{ads} of Ph-NO₂ on Cu_{SA}-Rh is beneficial for the rapid conversion of Ph-NO₂ to Ph-NOOH* by coupling with the protons in H₂O, and the unstable Ph-NOOH* rapidly removes OH* to further form the stable and important intermediate Ph-NO*, thus enhancing the rapid protonation in the Ph-NO₂ ERR process (Please see Ref. 8. ACS

Catal. 2022, 12, 14062-14071). Moreover, based on further analysis of charge density difference for adsorbed Ph-NO₂* on Cu_{SA}-Rh (111) and Rh (111), the electron transfer between Cu_{SA}-Rh and Ph-NO₂* (0.19 e) is smaller than that between Rh and Ph-NO₂* (0.25 e) owing to the electron interaction between Cu single-atom and the Rh host, which eventually can induce the decrease of electron interaction between Cu_{SA}-Rh and Ph-NO₂*, thus weakening the binding of Ph-NO₂* on Cu_{SA}-Rh (Please see Ref. 43. Adv. Funct. Mater. 2023, 33, 2209890), which is consistent with the results of adsorption energy. Secondly, there is electron accumulation in the antibonding orbitals for both Cu_{SA}-Rh and Rh, which favors the weakening of the bond energy for the N-O bond thus facilitating the conversion of Ph-NO₂* to key intermediates (Please see Ref. 43. Adv. Funct. Mater. 2023, 33, 2209890 and Ref. 56. Adv. Energy Mater. 2020, 10, 1903038). These conclusions indicate that the electron interaction between the isolated Cu single-atom and the Rh host causes a downshift of the *d*-band center and a decrease in the electron interaction between the catalyst and adsorbate, thus promoting a fast conversion of the reactants to key intermediates. Please see Page 16 and Ref. 8, 43, 56 and Figs. 6d and Figs. S26, S27, S28a and Table S4-S5.

Revision made in manuscript: Please see Page 16 and Ref. 8, 43, 56 and Figs. 6d. We have added the following revisions to the manuscript and highlighted them in the marked-up revised manuscript.

Moreover, as displayed in **Fig. 6d**, the *d*-band center of Rh-4d orbitals for Cu_{SA}-Rh (111) (-1.79 eV) exhibits a slight negative shift compared with Rh (111) (-1.77 eV). It is notable that the PDOS for Rh-4d of Cu_{SA}-Rh bulk (-1.82 eV) and Rh bulk (-1.49 eV) also reflect the similar trend (**Fig. S26**). As shown in **Fig. S27** and **Table S4**, The electron transfer between Cu_{SA}-Rh and Ph-NO₂* (0.19 e) is smaller than that between Rh and Ph-NO₂* (0.25 e) owing to the electron interaction between Cu single-atom and the Rh host, which eventually can induce the decrease of electron interaction between Cu_{SA}-Rh and Ph-NO₂*, thus weakening the binding of Ph-NO₂* on Cu_{SA}-Rh.⁴³ Notably, there is electron accumulation in the antibonding orbitals for both Cu_{SA}-Rh and Rh, which favors the weakening of the bond energy for the N-O bond.^{43,56} These conclusions indicate that the electron interaction between the isolated Cu single-atom and the Rh host causes a downshift of the *d*-band center and a decrease in the electron interaction between the catalyst and adsorbate, thus promoting a fast conversion of the reactants to key intermediates as well as optimizing the desorption of the target product.^{8,43} Moreover, **Fig. S28a** and **Table S5** present the optimized adsorption structure and adsorption energy (ΔE_{ads}) of Ph-NO₂ on Rh (111) and Cu_{SA}-Rh (111)

surfaces. The calculated Ph-NO₂ ΔE_{ads} of Cu_{SA}-Rh (111) is -1.08 eV, which is lower than that of Rh (111) (-1.27 eV). The weakened ΔE_{ads} of Ph-NO₂ on Cu_{SA}-Rh is beneficial for the rapid conversion of Ph-NO₂ to Ph-NOOH* by coupling with the protons in H₂O, and the unstable Ph-NOOH* rapidly removes OH* to further form the stable and important intermediate Ph-NO*, thus enhancing the rapid protonation in the Ph-NO₂ ERR process.⁸

8. Ma J, Wang Z, Majima T, Zhao G. Role of Ni in PtNi alloy for modulating the proton–electron transfer of electrocatalytic hydrogenation revealed by the in situ raman–rotating disk electrode method. *ACS Catal.* **12**, 14062-14071 (2022).

43. Chen K, Ma ZY, Li XC, Kang JL, Ma DW, Chu K. Single-Atom Bi Alloyed Pd Metallene for Nitrate Electroreduction to Ammonia. *Adv. Funct. Mater.* **33**, 2209890 (2023).

56. Cao D, Xu H, Cheng D. Construction of defect-rich rhcu nanotubes with highly active Rh₃Cu₁ alloy phase for overall water splitting in all pH values. *Adv. Energy Mater.* **10**, 1903038 (2020).

Fig. 6. (d) The PDOSs for Rh-4d of Rh (111) and CuSA-Rh (111) surfaces.

Revision made in supporting information: Please see Pages S16, S17, S23, S24 and Figs. S26, S27, S28a and Table S4-S5. We have added the following revisions to the supporting information.

Fig. S26. The PDOSs for Rh-4d of Rh bulk and Cu_{SA} -Rh bulk.

Fig. S27. Charge density difference for adsorbed Ph-NO_2^* on (a) Cu_{SA} -Rh (111) and (b) Rh (111). The blue and red spheres represent Rh and Cu atoms respectively, as well as the yellow and cyan indicate the charge depletion and accumulation areas.

Fig. S28. Calculated adsorption energies of (a) Ph-NO_2^* and (b) Ph-NO^* on Rh (111) and Cu_{SA} -Rh (111).

Comment 2-6: The most important one is, besides the adsorption energy calculations, the whole free energy profiles (including water dissociation) should be provided to support their proposed mechanisms of the two reactions. Detailed comparison should be done between those with and without the Cu. All the related data leading to these energy profiles should be summarized in supporting information.

Reply 2-6: Thanks for the value suggestion. We have provided the whole free energy profiles of the two reactions and the energy profiles of the H₂O dissociation process to illustrate the enhanced mechanism of the electrocatalytic reaction. All the related data leading to these energy profiles have been summarized in supporting information. For Ph-NO₂ ERR, the free energy profiles of optimized intermediates for Ph-NO₂ ERR pathway reveal Ph-NOH*→Ph-NHOH* and Ph-NH₂*→Ph-NH₂ as the rate-determining step (RDS) for Cu_{SA}-Rh and Rh, respectively. Obviously, the Cu_{SA}-Rh exhibits a lower energy barrier (0.74 eV) on the RDS compared with Rh (0.85 eV), further indicating the higher ability on Cu_{SA}-Rh for driving the Ph-NO₂ ERR to Ph-NH₂. For MOR, it can be observed that the Cu_{SA}-Rh possesses a lower energy barrier for RDS (CH₂O*→CHO*, 0.55 eV) compared to the RDS (CHO*→HCOOH*, 0.62 eV) of Rh, indicating a more favorable MOR process on Cu_{SA}-Rh. Moreover, the energy barrier for the desorption step (HCOOH*→HCOOH) of Cu_{SA}-Rh (0.43 eV) is lower than that of Rh (0.54 eV), indicating the easier desorption of the target product from the Cu_{SA}-Rh MAS surface, which is conducive to facilitating the HCOO⁻ formation. For H₂O dissociation process, the energy barrier for H₂O dissociation of Cu_{SA}-Rh (0.83 eV) is lower than that of Rh (1.01 eV), indicating that the introduction of the Cu single-atom is beneficial for the dissociation of H₂O on Cu_{SA}-Rh to facilitate the formation H* for Ph-NO₂ hydrogenation. Please see Pages 17, 18 and Fig. 6e, 6f, 6g, 6j, 6k and Fig. S29, S30, S32 and Table S3-S10.

Revision made in manuscript: Please see Pages 17, 18 and Fig. 6e, 6f, 6g, 6j, 6k. We have added the following revisions to the manuscript and highlighted them in the marked-up revised manuscript.

Moreover, the free energy profiles of optimized intermediates for Ph-NO₂ ERR pathway reveal Ph-NOH*→Ph-NHOH* and Ph-NH₂*→Ph-NH₂ as the rate-determining step (RDS) for Cu_{SA}-Rh and Rh, respectively (Figs 6e-6f and Fig. S29). Obviously, the Cu_{SA}-Rh exhibits a lower energy barrier (0.74 eV) on the RDS compared with Rh (0.85 eV) (Fig. 6f and Table S6), further indicating the higher ability on Cu_{SA}-Rh for driving the Ph-NO₂ ERR to Ph-NH₂.

Fig. 6g and Fig. S30 display the calculated energy profiles of H₂O dissociation process and corresponding optimized structures of the initial, transition and final states. The energy barrier for H₂O dissociation of Cu_{SA}-Rh (0.83 eV) is lower than that of Rh (1.01 eV) (Fig. 6g and Table S7), indicating that the introduction of the Cu single-atom is beneficial for the dissociation of H₂O on Cu_{SA}-Rh to facilitate the formation H* for Ph-NO₂ hydrogenation.

For further investigating the MOR mechanism on $\text{Cu}_{\text{SA}}\text{-Rh}$ MAs, the free energy profiles of optimized intermediates for MOR pathway were analyzed by the DFT calculations (**Fig. 6j-6k and Fig. S32**). It can be observed that the $\text{Cu}_{\text{SA}}\text{-Rh}$ possesses a lower energy barrier for RDS ($\text{CH}_2\text{O}^* \rightarrow \text{CHO}^*$, 0.55 eV) compared to the RDS ($\text{CHO}^* \rightarrow \text{HCOOH}^*$, 0.62 eV) of Rh (**Fig. 6k and Table S9**), indicating a more favorable MOR process on $\text{Cu}_{\text{SA}}\text{-Rh}$. Meanwhile, **Fig. 6k** further shows that the energy barrier for the desorption step ($\text{HCOOH}^* \rightarrow \text{HCOOH}$) of $\text{Cu}_{\text{SA}}\text{-Rh}$ (0.43 eV) is lower than that of Rh (0.54 eV).

Fig. 6. (e) Optimized structures of Ph-NO₂ ERR intermediates on $\text{Cu}_{\text{SA}}\text{-Rh}$ (111). The blue, red, gray, cyan, purple, and white spheres represent Rh, Cu, C, O, N, and H atoms respectively. (f) Comparison of free energy profiles for Ph-NO₂ ERR pathway on $\text{Cu}_{\text{SA}}\text{-Rh}$ (111) and Rh (111). (g) Calculated energy profiles of H₂O dissociation on $\text{Cu}_{\text{SA}}\text{-Rh}$ (111) and Rh (111). (h) Optimized structures of adsorbed H* and (i) calculated H* adsorption energies variations at different sites on $\text{Cu}_{\text{SA}}\text{-Rh}$ (111). The blue, red, and white spheres represent Rh, Cu, and H atoms respectively. (j) Optimized structures of MOR intermediates on $\text{Cu}_{\text{SA}}\text{-Rh}$ (111). The blue, red, gray, cyan, and white spheres represent Rh, Cu, C, O, and H atoms respectively. (k) Comparison of free energy profiles for MOR pathway on $\text{Cu}_{\text{SA}}\text{-Rh}$ (111) and Rh (111).

Revision made in supporting information: Please see Pages S17, S18 and Pages S22-S29 and Figs. S29, S30, S32 and Table S3-S10. We have added the following revisions to the supporting information.

Fig. S29. Optimized structures of Ph-NO₂ ERR intermediates on Rh (111). The blue, gray, cyan, purple, and white spheres represent Rh, C, O, N, and H atoms respectively.

Fig. S30. Optimized structures for the initial, transition and final states of H₂O dissociation on Rh (111). The blue, cyan, and white spheres represent Rh, O, and H atoms respectively.

Fig. S32. Optimized structures of MOR intermediates on Rh (111). The blue, gray, cyan, and white spheres represent Rh, C, O, and H atoms respectively.

Reviewer #3

Comment: In this paper, the authors synthesized Cu single-atom dispersed Rh metallene arrays on Cu foam and use them as electrocatalysts for the reduction of nitrobenzene and oxidation of methanol. Characterization of the catalyst was carried out using a combination of several techniques. The electrochemical reaction yielded the target products (aniline and formate) with high selectivity. A reaction mechanism is proposed using DFT calculations. The reviewer carefully considered the paper. The publication in Nature Communications would require greater scholarly significance than in previous studies. However, the synthesis of aniline by electro-oxidation of nitrobenzene and the synthesis of formic acid by oxidation of methanol have already been reported; thus, there is no novelty in the reactions. Although there may not be any studies combining these two reactions, as an electrolysis process, it is merely a combination of two known reactions. The remaining novelty may lie in the performance of the catalyst. However, there are still doubts regarding the significance of the catalytic activity. While Table S2 compares the results with previous studies, it is important to note that the electrochemical reactions being compared are fundamentally different. Therefore, comparing voltages among different reactions does not hold significant meaning. Additionally, it should be noted that the majority of catalysts in previous studies were transition metal catalysts, while this study utilizes noble metals (which generally exhibit higher catalytic activity). Hence, making such comparisons may not be meaningful in this context. For the above reasons, the reviewer has determined that this paper is not acceptable for publication in Nature Communications. Other comments are noted below.

Reply: We thank Referee #3 for his/her valuable time reviewing our manuscript. We also appreciate his/her comments. Although Ph-NO₂ ERR and MOR have been reported so far, in this work, we constructed for the first time Cu single-atom coordinated Rh metallene array structure and proposed the unique mechanisms (H*-spillover effect and synergistic catalytic effect) to enhance the activity and selectivity of Ph-NO₂ ERR and MOR, which is different from the previously reported work. More importantly, we constructed the Cu_{SA}-Rh MAs/CF as a bifunctional catalyst applied in a novel Ph-NO₂ ERR-MOR two-electrode system, which achieves the simultaneous generation of high-value chemicals at the cathode and anode. Meanwhile, we proved the unique mechanism of the introduction of Cu single-atom on the enhancement of catalytic activity by a series of experiments and DFT calculations. The design of this single-atom alloy metallene structure and the use of H*-

spillover effect and synergistic catalytic effect are promising strategies for applying in other organic electrocatalytic conversion reactions. Thus, this work is innovative. Moreover, your comments on the catalytic performance comparison table are extremely important, we have removed the previous Table S2 and developed a new table. Based on the new Table S2, apart from comparing noble metal catalysts, we also further compared non-noble metal catalysts. Although the designed catalyst ($\text{Cu}_{\text{SA}}\text{-Rh MAs/CF}$) in this work is a noble metal-based catalyst, due to the synergistic catalytic effect and H^* -spillover effect caused by the interaction of Cu single-atom and Rh host, the $\text{Cu}_{\text{SA}}\text{-Rh MAs/CF}$ exhibits a lower electrolysis potential (-0.1 V vs. RHE) and electrolysis time (1 h) to achieve a higher Ph- NO_2 conversion ($\sim 100\%$) and Ph- NH_2 selectivity (99.7%), which is significantly higher than previously reported noble metal based/non-noble metal-based catalysts. Please see Page 11 and Table S2.

Revision made in manuscript: Please see Page 11. We have added the following revisions to the manuscript and highlighted them in the marked-up revised manuscript.

Notably, the activity and selectivity of Ph- NO_2 ERR to Ph- NH_2 on $\text{Cu}_{\text{SA}}\text{-Rh MAs/CF}$ are also superior to several reported electrocatalysts (Table S2).

Revision made in supporting information: Please see Page S21 and Table S2.

Table S2. Ph- NO_2 ERR performance of $\text{Cu}_{\text{SA}}\text{-Rh MAs/CF}$ compared with other reported catalysts.

catalyst	Potential	time (h)	Ph- NO_2 Conversion (%)	Ph- NH_2 Selectivity (%)	Ref.
$\text{Cu}_{\text{SA}}\text{-Rh MAs/CF}$	-0.1 V vs. RHE	1	~ 100	99.7	/
Au@PtNi	-0.2 V vs. RHE	2	98.1	81.8	5
Au@Pt	-0.2 V vs. RHE	2	82.8	64.1	5
$\text{Co}_3\text{S}_{4-x}$ NS	$-1.0 \text{ V vs. Hg/HgO}$	6	~ 98.0	~ 99.0	6
$\text{Cu}_3\text{Pt/C}$	-0.3 V vs. RHE	/	~ 100	~ 99.0	7
TiO_{2-x} single crystal	-0.55 V vs. RHE	2	~ 100	72.0	8
Cu/Ti	-0.29 V vs. RHE	2.5	93.9	97.1	9

Layered iron(II)-iron(III) hydroxides	-0.5 V vs. Ag/AgCl	5.5	74.5	37.8	10
--------------------	-----	------	------	----

Comment 3-1: The effect of Cu single-atom also seems unclear; the effect of Cu single-atom cannot be discussed without preparing Rh metallene arrays without Cu single-atom and comparing their catalytic activity. The comparison with Rh nanoparticles is made in the paper, but as mentioned in the paper, the surface areas are different between Rh NPs and Rh metallene. Therefore, the comparison may not be appropriate for discussing the effect of Cu single atoms due to the differences in surface area.

Reply 3-1: Thanks for the value suggestion. We have supplied the catalytic performance of Rh metallene-CF (Rhene-CF) and demonstrated the effect of the introduction of Cu single-atom on catalytic performance by comparing Cu_{SA}-Rh MAs/CF with a series of comparison samples. Firstly, for cathodic Ph-NO₂ ERR, LSV curves for various electrocatalysts demonstrate that the Cu_{SA}-Rh MAs/CF possesses a stronger electroreduction activity compared with Rhene-CF, Rh nanoparticles-CF (Rh NPs-CF) and CF, and the Ph-NH₂ selectivity (99.7%) of Cu_{SA}-Rh MAs/CF is much higher than those of Rhene-CF (64.7%), Rh NPs-CF (59.3%) and CF (28.3%). With respect to the standard potential (0.89 V vs. RHE) of the Ph-NO₂ ERR (Please see Ref. 21. Appl. Catal. B: Environ. 2018, 226, 509-522), the overpotential (0.532 V vs. RHE) of Cu_{SA}-Rh MAs/CF is lower than those of Rhene-CF (0.667 V vs. RHE), Rh NPs-CF (0.619 V vs. RHE) and CF (0.638 V vs. RHE) for achieving a current density of -50 mA cm⁻². Secondly, for anodic MOR, the Cu_{SA}-Rh MAs/CF possesses a superior MOR activity compared with Rhene-CF, Rh NPs-CF and CF. Relative to the standard potential (0.103 V vs. RHE) of MOR (Please see Ref. 17. Nat. Commun. 2023, 14, 1686), the overpotential of Cu_{SA}-Rh MAs/CF is 1.25 V (vs. RHE) for reaching a current density of 20 mA cm⁻², which is lower than those of Rhene-CF (1.30 V vs. RHE), Rh NPs-CF (1.31 V vs. RHE) and CF (1.35 V vs. RHE) Thirdly, As revealed by EIS for various electrocatalysts, the Cu_{SA}-Rh MAs/CF (1.13 Ω) presents a smaller resistance (*R*_{ct}) value than Rhene-CF (1.96 Ω), Rh NPs-CF (3.34 Ω) and CF (9.68 Ω), revealing a rapid interfacial charge transfer on Cu_{SA}-Rh MAs/CF. Fourthly, the ECSAs for various electrocatalysts were evaluated by the electrochemical *C*_{dl} calculated based on CV curves, the *C*_{dl} value for Cu_{SA}-Rh MAs/CF (49.9 mF cm⁻²) was calculated to be higher than those of Rhene-CF (35.9 mF cm⁻²), Rh NPs-CF (17.1 mF cm⁻²) and CF (2.1 mF cm⁻²), which reveals the rich active sites in Cu_{SA}-Rh MAs/CF due to the ultrathin metallene array structure and

the introduction of isolated Cu single-atom. These conclusions demonstrate that the ultrathin metallene structure and the introduction of isolated Cu single-atom are beneficial for improving Ph-NO₂ ERR and MOR activity. Please see Page 11, 13, 14 and Figs. S14, S15, S17, S18, S19, S20.

Revision made in manuscript: Please see Pages 11, 13, 14. We have added the following revisions to the manuscript and highlighted them in the marked-up revised manuscript.

LSV curves for various electrocatalysts demonstrate that the Cu_{SA}-Rh MAS/CF possesses a stronger electroreduction activity compared with Rh metallene-CF (Rhene-CF), Rh nanoparticles-CF (Rh NPs-CF) and CF (**Fig. S14a**), and the Ph-NH₂ selectivity (99.7%) of Cu_{SA}-Rh MAS/CF is much higher than those of Rhene-CF (64.7%), Rh NPs-CF (59.3%) and CF (28.3%) (**Fig. S14b**). Moreover, with respect to the standard potential (0.89 V *vs.* RHE) of the Ph-NO₂ ERR,²¹ the overpotential (0.532 V *vs.* RHE) of Cu_{SA}-Rh MAS/CF is lower than those of Rhene-CF (0.667 V *vs.* RHE), Rh NPs-CF (0.619 V *vs.* RHE) and CF (0.638 V *vs.* RHE) for achieving a current density of -50 mA cm⁻² (**Fig. S15**), further suggesting a superior Ph-NO₂ ERR activity on Cu_{SA}-Rh MAS/CF. The improved activity of Ph-NO₂ ERR to Ph-NH₂ for Cu_{SA}-Rh MAS/CF originates from the ultrathin metallene array structure and the synergistic effect of isolated Cu single-atom with Rh host. It is worth mentioning that the Cu_{SA}-Rh MAS/CF possesses a superior MOR activity compared with Rhene-CF, Rh NPs-CF and CF (**Fig. S17a**). Meanwhile, relative to the standard potential (0.103 V *vs.* RHE) of MOR,¹⁷ the overpotential of Cu_{SA}-Rh MAS/CF is 1.25 V (*vs.* RHE) for reaching a current density of 20 mA cm⁻², which is lower than those of Rhene-CF (1.30 V *vs.* RHE), Rh NPs-CF (1.31 V *vs.* RHE) and CF (1.35 V *vs.* RHE) (**Fig. S17b**).

As displayed in **Fig. S18**, the Cu_{SA}-Rh MAS/CF (1.13 Ω) presents a smaller resistance (R_{ct}) value than Rhene-CF (1.96 Ω), Rh NPs-CF (3.34 Ω) and CF (9.68 Ω), revealing a rapid interfacial charge transfer on Cu_{SA}-Rh MAS/CF.^{52,53} Besides, the electrochemical active surface areas (ECSAs) for various electrocatalysts were evaluated by the electrochemical double-layer capacitance (C_{dl}) calculated based on cyclic voltammetry (CV) curves (**Fig. S19**). The C_{dl} value for Cu_{SA}-Rh MAS/CF (49.9 mF cm⁻²) was calculated to be higher than those of Rhene-CF (35.9 mF cm⁻²), Rh NPs-CF (17.1 mF cm⁻²) and CF (2.1 mF cm⁻²) (**Fig. S20**), which reveals the rich active sites in Cu_{SA}-Rh MAS/CF due to the ultrathin metallene array structure and the introduction of isolated Cu single-atom.

17. Zhu B, *et al.* Unraveling a bifunctional mechanism for methanol-to-formate electro-oxidation

on nickel-based hydroxides. *Nat. Commun.* **14**, 1686 (2023).

21. Daems N, *et al.* Selective reduction of nitrobenzene to aniline over electrocatalysts based on nitrogen-doped carbons containing non-noble metals. *Appl. Catal. B: Environ.* **226**, 509-522 (2018).

Revision made in supporting information: Please see Page S11-S14 and Figs. S14, S15, S17, S18, S19, S20. We have added the following revisions to the supporting information.

Fig. S14. (a) LSV curves of Cu_{SA}-Rh MAS/CF, Rhene-CF, Rh NPs-CF and CF in 1 M KOH + 5 mM Ph-NO₂ solutions. (b) Ph-NO₂ conversion and Ph-NH₂ selectivity of Cu_{SA}-Rh MAS/CF, Rhene-CF, Rh NPs-CF and CF at -0.1 V (vs. RHE).

Fig. S15. Comparison of the overpotential for Cu_{SA}-Rh MAS/CF, Rhene-CF, Rh NPs-CF and CF relative to the standard potential of the Ph-NO₂ ERR at a current density of -50 mA cm⁻²

Fig. S17. (a) LSV curves of Cu_{SA}-Rh MAS/CF, Rhene-CF, Rh NPs-CF and CF in 1 M KOH + 4 M CH₃OH solutions. (b) Comparison of the overpotential for Cu_{SA}-Rh MAS/CF, Rhene-CF, Rh NPs-CF and CF relative to the standard potential of the MOR at a current density of 20 mA cm⁻².

Fig. S18. EIS plots of Cu_{SA}-Rh MAS/CF, Rhene-CF, Rh NPs-CF and CF in 1 M KOH solutions at applied potentials: -0.1 V (vs. RHE).

Fig. S19. CVs of (a) Cu_{SA}-Rh MAS/CF, (b) Rhene-CF, (c) Rh NPs-CF and (d) CF in the region of

0.404-0.504 V (vs. RHE).

Fig. S20. Capacitive current densities at 0.454 V (vs. RHE) derived from CVs against scan rates of Cu_{SA}-Rh MAs/CF, Rhene-CF, Rh NPs-CF and CF.

Comment 3-2: The authors stated the electronic interaction between Cu and Rh, but what is the density of Cu in the Rh metallene? If Cu is sparsely present on Rh metallene, there would be Rh atoms that do not interact with Cu. What is the percentage of Rh that the effect of Cu does reach?

Reply 3-2: Thanks for the valuable question. It is extremely difficult to provide specific Cu intensity values and percentages of Cu-Rh effect. However, we can analyze the effect of Cu single atoms through a series of experiments and DFT calculations. The electron interaction between Cu single-atom and Rh host originates from the electron transfer between Cu and Rh. Bader charge calculation analysis reveals that the electron transfer is from the Cu single-atom to the Rh host. Charge density difference calculation analysis of Cu_{SA}-Rh reveals the accumulation of negative charge around the Rh host while Cu single-atom possesses positive charge property due to the loss of charge, which further indicate the electron-rich nature of Rh localization resulting from the introduction of isolated Cu single-atom. Visually, based on the Bader charge and charge density difference graphs of Cu_{SA}-Rh, although the content of Cu single-atom in the Cu_{SA}-Rh model is very small, this has an impact on the electron distribution around the Rh atoms. Moreover, the introduction of Cu single-atom has an impact on the adsorption energies of different Rh sites by the calculated H* adsorption energies of different sites, which indicates a H*-spillover process between the isolated Cu single-atom and the Rh host. The electron interaction between Cu and Rh also

affects the *d*-band center of Rh. Meanwhile, based on the elemental mapping image selected from random regions, the distribution of Cu single-atom is homogeneous on Rh metallene. Therefore, the influence range of Cu single-atom as highly reactive atoms on Rh metallene is significant and homogeneous, which fits the advantages of single-atom alloys (Please see Ref. 25. *Adv. Energy Mater.* 2022, 12, 2201823 and Ref. 26. *Chem. Rev.* 2020, 120, 12044-12088 and Ref. 43 *Adv. Funct. Mater.* 2023, 33, 2209890). Please see Pages 6, 15, 16, 17 and Ref. 25, 26, 43 and Figs. 6a, 6b, 6d, 6h, 6i and Figs. S24-S26 and Table S3 and Table S8.

Revision made in manuscript: Please see Pages 6, 15, 16, 17 and Ref. 25, 26, 43 and Figs. 6a, 6b, 6d, 6h, 6i. We have added the following revisions to the manuscript and highlighted them in the marked-up revised manuscript.

Fig. 1d presents the aberration-corrected high angle annular dark field scanning transmission electron microscopy (AC-HAADF-STEM) and corresponding elemental mapping images of Cu_{SA}-Rh MAs, revealing the homogeneous distribution of Cu atoms in Cu_{SA}-Rh MAs.

Based on the Bader charge calculation analysis (**Fig. S24 and Table S3**), the total net charges of Rh and Cu are 1.94 e and -1.94 e, respectively, which reveals that the electron transfer is from the Cu single-atom to the Rh host. **Figs. 6a-6b and Fig. S25** reveal the accumulation of negative charge around the Rh host while Cu single-atom possesses positive charge property due to the loss of charge, which further indicate the electron-rich nature of Rh localization resulting from the introduction of isolated Cu single-atom.

Moreover, as displayed in **Fig. 6d**, the *d*-band center of Rh-4d orbitals for Cu_{SA}-Rh (111) (-1.79 eV) exhibits a slight negative shift compared with Rh (111) (-1.77 eV). It is notable that the PDOS for Rh-4d of Cu_{SA}-Rh bulk (-1.82 eV) and Rh bulk (-1.49 eV) also reflect the similar trend (**Fig. S26**).

Furthermore, the H* ΔE_{ads} of various sites (A to E) were further calculated for investigating the hydrogenation mechanism of H* with H₂O as the hydrogen source on Cu_{SA}-Rh MAs (**Figs. 6h-6i**).

Visually, the variation of H* ΔE_{ads} from A to E sites on Cu_{SA}-Rh reveals a gradually increased H* adsorption (**Fig.6i and Table S8**), which indicates a H*-spillover process between the isolated Cu single-atom and the Rh host.

25. Shen T, Wang S, Zhao TH, Hu YZ, Wang DL. Recent advances of single-atom-alloy for energy electrocatalysis. *Adv. Energy Mater.* **12**, 2201823 (2022).

26. Hannagan RT, Giannakakis G, Flytzani-Stephanopoulos M, Sykes ECH. Single-atom alloy

catalysis. *Chem. Rev.* **120**, 12044-12088 (2020).

43. Chen K, Ma ZY, Li XC, Kang JL, Ma DW, Chu K. Single-Atom Bi Alloyed Pd Metallene for Nitrate Electroreduction to Ammonia. *Adv. Funct. Mater.* **33**, 2209890 (2023).

Fig. 6. (a) Charge density difference on $\text{Cu}_{\text{SA}}\text{-Rh}$ model. The blue and red spheres represent Rh and Cu atoms respectively, as well as the yellow and cyan indicate the charge depletion and accumulation areas. (b) Slice of charge density difference for $\text{Cu}_{\text{SA}}\text{-Rh}$ bulk structure. (c) The PDOSs of $\text{Cu}_{\text{SA}}\text{-Rh}$. (d) The PDOSs for Rh-4d of Rh (111) and $\text{Cu}_{\text{SA}}\text{-Rh}$ (111) surfaces. (e) Optimized structures of Ph- NO_2 ERR intermediates on $\text{Cu}_{\text{SA}}\text{-Rh}$ (111). The blue, red, gray, cyan, purple, and white spheres represent Rh, Cu, C, O, N, and H atoms respectively. (f) Comparison of free energy profiles for Ph- NO_2 ERR pathway on $\text{Cu}_{\text{SA}}\text{-Rh}$ (111) and Rh (111). (g) Calculated energy profiles of H_2O dissociation on $\text{Cu}_{\text{SA}}\text{-Rh}$ (111) and Rh (111). (h) Optimized structures of adsorbed H^* and (i) calculated H^* adsorption energies variations at different sites on $\text{Cu}_{\text{SA}}\text{-Rh}$ (111). The blue, red, and white spheres represent Rh, Cu, and H atoms respectively.

Revision made in supporting information: Please see Page S16 and Figs. S24-S26 and Table S3 and Table S8. We have added the following revisions to the supporting information.

Fig. S24. (a) Net charge obtained from Bader charge analysis marked on Rh (blue balls) and Cu atoms (red balls). (b) Bader charge of Cu_{SA}-Rh (the balls marked by red circles represent the tendency for electron transfer on Cu single-atom and the unmarked balls represent the tendency for electron transfer on Rh host).

Fig. S25. (a) Main view, (b) side view and (c) top view of charge density difference on Cu_{SA}-Rh. The blue and red spheres represent Rh and Cu atoms respectively, as well as the yellow and cyan indicate the charge depletion and accumulation areas.

Fig. S26. The PDOSs for Rh-4d of Rh bulk and Cu_{SA}-Rh bulk.

Comment 3-3: There is a lack of discussion from an electrochemical point of view. At the very least, the standard redox potential should be given and overvoltages should be discussed.

Reply 3-3: Thanks for the value suggestion. We have discussed the standard redox potential and

overpotential in detail. For Ph-NO₂ ERR, with respect to the standard potential (0.89 V vs. RHE) of the Ph-NO₂ ERR (Please see Ref. 21. *Appl. Catal. B: Environ.* 2018, 226, 509-522), the overpotential (0.532 V vs. RHE) of Cu_{SA}-Rh MAs/CF is lower than those of Rhene-CF (0.667 V vs. RHE), Rh NPs-CF (0.619 V vs. RHE) and CF (0.638 V vs. RHE) for achieving a current density of -50 mA cm⁻², further suggesting a superior Ph-NO₂ ERR activity on Cu_{SA}-Rh MAs/CF. For MOR, relative to the standard potential (0.103 V vs. RHE) of MOR (Please see Ref. 17. *Nat. Commun.* 2023, 14, 1686), the overpotential of Cu_{SA}-Rh MAs/CF is 1.25 V (vs. RHE) for reaching a current density of 20 mA cm⁻², which is lower than those of Rhene-CF (1.30 V vs. RHE), Rh NPs-CF (1.31 V vs. RHE) and CF (1.35 V vs. RHE). Please see Pages 11, 13 and Ref. 17, 21 and Figs. S15, S17.

Revision made in manuscript: Please see Pages 11, 13. We have added the following revisions to the manuscript and highlighted them in the marked-up revised manuscript.

Moreover, with respect to the standard potential (0.89 V vs. RHE) of the Ph-NO₂ ERR,²¹ the overpotential (0.532 V vs. RHE) of Cu_{SA}-Rh MAs/CF is lower than those of Rhene-CF (0.667 V vs. RHE), Rh NPs-CF (0.619 V vs. RHE) and CF (0.638 V vs. RHE) for achieving a current density of -50 mA cm⁻² (**Fig. S15**), further suggesting a superior Ph-NO₂ ERR activity on Cu_{SA}-Rh MAs/CF. The improved activity of Ph-NO₂ ERR to Ph-NH₂ for Cu_{SA}-Rh MAs/CF originates from the ultrathin metallene array structure and the synergistic effect of isolated Cu single-atom with Rh host. Meanwhile, relative to the standard potential (0.103 V vs. RHE) of MOR,¹⁷ the overpotential of Cu_{SA}-Rh MAs/CF is 1.25 V (vs. RHE) for reaching a current density of 20 mA cm⁻², which is lower than those of Rhene-CF (1.30 V vs. RHE), Rh NPs-CF (1.31 V vs. RHE) and CF (1.35 V vs. RHE) (**Fig. S17b**).

17. Zhu B, *et al.* Unraveling a bifunctional mechanism for methanol-to-formate electro-oxidation on nickel-based hydroxides. *Nat. Commun.* **14**, 1686 (2023).

21. Daems N, *et al.* Selective reduction of nitrobenzene to aniline over electrocatalysts based on nitrogen-doped carbons containing non-noble metals. *Appl. Catal. B: Environ.* **226**, 509-522 (2018).

Revision made in supporting information: Please see Pages S11-S112 and Figs. S15, S17. We have added the following revisions to the supporting information.

Fig. S15. Comparison of the overpotential for Cu_{SA}-Rh MAs/CF, Rhene-CF, Rh NPs-CF and CF relative to the standard potential of the Ph-NO₂ ERR at a current density of -50 mA cm⁻².

Fig. S17. (a) LSV curves of Cu_{SA}-Rh MAs/CF, Rhene-CF, Rh NPs-CF and CF in 1 M KOH + 4 M CH₃OH solutions. (b) Comparison of the overpotential for Cu_{SA}-Rh MAs/CF, Rhene-CF, Rh NPs-CF and CF relative to the standard potential of the MOR at a current density of 20 mA cm⁻².

Comment 3-4: Finally, the authors would need to check references. In the 52 references, only 4 papers are published by different nationalities than the authors. If appropriate works related to this study are cited, the reference list is no problem, but the reviewer is concerned about the large nationality bias in the reference list.

Reply 3-4: Thanks for your value suggestion. All of the cited references are very relevant to this work. No any nationality bias exists in the citation of the references. Anyway, we have tried our best to reconsider and cite the relevant references to avoid this misunderstanding. Please see References in Pages 23-27.

REVIEWERS' COMMENTS

Reviewer #1 (Remarks to the Author):

All the comments and suggestions from the reviewers have been replied very well. The quality of the revised manuscript is satisfied. I recommend its acceptance in Nature Communications.

Reviewer #2 (Remarks to the Author):

The authors have tried to address my previous concerns about the insufficient computational evidence. The current version with more details is clearly much better and can provide a stronger support for their proposed possible reaction mechanism resulting in the good catalytic performance of synthesized catalysts.

Reviewer #3 (Remarks to the Author):

I have evaluated the authors' responses. However, the major concern I raised, which is whether this paper has the novelty to be published in Nature Communications, is still not clear.

Firstly, as the authors themselves acknowledge, the reactions covered in this paper are well-known and are simply combined here. While the authors claim there are some novelties related to DFT calculations, I cannot evaluate them due to not being within my expertise, so I will leave that aside. However, since the main focus of this study is electrochemical reactions, innovative performance in these reactions must be needed for publication in Nature Communications. With this in mind, I examined the revised Table S2. In Table S2, a comparison between the hydrogenation of nitrobenzene in this paper and previous studies is provided.

I have reviewed all the papers shown in Table S2. As I mentioned in my previous comments, it is not appropriate to compare precious metal catalysts and transition metal catalysts on the same level. Furthermore, it is essential to emphasize that the reaction conditions differ between previous papers and this work, making a direct comparison of potential, conversion, and selectivity meaningless. For example, in ref 5, the reaction is performed with 0.1 M nitrobenzene, which is significantly higher than your paper (5 mM). Additionally, the electrolyte used is not consistent. While you used KOH, previous studies often employed sodium sulfate and others. Furthermore, the amount of catalysts on an electrode would be different. In other words, comparing potential, conversion, and selectivity in reactions with entirely different pH, substrate concentrations, and ESCA of catalysts does not prove the superiority of your research.

Moreover, most of the papers listed in Table S2 are not from high-impact journals. If you wish to emphasize excellent catalytic performance in Nature Communications, it is also crucial to consider what the comparison papers are.

Based on the points mentioned above, due to the lack of clarity on the significance of this paper's electrochemical reactions, I cannot approve its publication in Nature Communications.

Point-to-Point Response to Reviewers

Reviewer #1

Comment: All the comments and suggestions from the reviewers have been replied very well. The quality of the revised manuscript is satisfied. I recommend its acceptance in Nature Communications.

Reply: We thank Referee #1 for his/her valuable time reviewing our manuscript. We also appreciate his/her positive comments and the recommendation for publishing.

Reviewer #2

Comment: The authors have tried to address my previous concerns about the insufficient computational evidence. The current version with more details is clearly much better and can provide a stronger support for their proposed possible reaction mechanism resulting in the good catalytic performance of synthesized catalysts.

Reply: We thank Referee #2 for his/her valuable time reviewing our manuscript. We also appreciate his/her positive comments and the recommendation for publishing.

Reviewer #3

Comment: I have evaluated the authors' responses. However, the major concern I raised, which is whether this paper has the novelty to be published in Nature Communications, is still not clear. Firstly, as the authors themselves acknowledge, the reactions covered in this paper are well-known and are simply combined here. While the authors claim there are some novelties related to DFT calculations, I cannot evaluate them due to not being within my expertise, so I will leave that aside. However, since the main focus of this study is electrochemical reactions, innovative performance in these reactions must be needed for publication in Nature Communications. With this in mind, I examined the revised Table S2. In Table S2, a comparison between the hydrogenation of nitrobenzene in this paper and previous studies is provided. I have reviewed all the papers shown in Table S2. As I mentioned in my previous comments, it is not appropriate to compare precious metal catalysts and transition metal catalysts on the same level. Furthermore, it is essential to emphasize that the reaction conditions differ between previous papers and this work, making a direct comparison of potential, conversion, and selectivity meaningless. For example, in ref 5, the reaction is performed with 0.1 M nitrobenzene, which is significantly higher than your paper (5 mM). Additionally, the electrolyte used is not consistent. While you used KOH, previous studies often employed sodium sulfate and others. Furthermore, the amount of catalysts on an electrode would be different. In other words, comparing potential, conversion, and selectivity in reactions with entirely different pH, substrate concentrations, and ESCA of catalysts does not prove the superiority of your research. Moreover, most of the papers listed in Table S2 are not from high-impact journals. If you wish to emphasize excellent catalytic performance in Nature Communications, it is also crucial to consider what the comparison papers are. Based on the points mentioned above, due to the lack of clarity on the significance of this paper's electrochemical reactions, I cannot approve its publication in Nature Communications.

Reply: We thank Referee #3 for his/her valuable time reviewing our manuscript. We also appreciate his/her comments. Your comments on the catalytic performance comparison table are extremely important, we have removed the Table S2 in the revised manuscript based on your suggestions. Moreover, as previously mentioned, although Ph-NO₂ ERR and MOR have been reported so far, there are some important distinctions that make our work different from previously reported work and contribute to its novelty. Firstly, we constructed for the first time Cu single-atom coordinated

Rh metallene array structure by a facile one-step solvothermal method, which belongs to the novel two-dimensional single-atom alloy catalysts. Benefiting from the unique metallene array structure, the stable security wall-like structure formed by the ultrathin metallene arrays provides sufficient active sites and abundant interlayer channels (Please see Ref. 45. ACS Sustainable Chem. Eng. 2019, 7, 10035-10043), and the inherent defect-rich structure and low-crystalline regions of Cu_{SA}-Rh MAs/CF can induce unsaturated coordination metallic bonds and optimize the local electron structure (Please see Ref. 37. Chem. Soc. Rev. 2021, 50, 6700-6719 and Ref. 56. Adv. Energy Mater. 2020, 10, 1903038). Secondly, we constructed the Cu_{SA}-Rh MAs/CF as a bifunctional catalyst applied in a novel Ph-NO₂ ERR-MOR two-electrode system, which achieves the simultaneous generation of high-value chemicals at the cathode and anode. Thirdly, we proposed the unique mechanisms (H*-spillover effect and synergistic catalytic effect) to enhance the activity and selectivity of Ph-NO₂ ERR and MOR, which is different from the previously reported work. More importantly, we proved the unique mechanism of the introduction of Cu single-atom on the enhancement of catalytic activity by a series of experiments and DFT calculations. The synergistic catalysis effect and H*-spillover effect between Cu single-atom and Rh host can optimize the catalytic reaction process, facilitate the stable and rapid conversion of reactants to intermediates as well as accelerate the desorption of target products, and the Cu single-atom as effective adsorption sites can modulate the competition for adsorbate adsorption on Rh sites thus promoting electrocatalytic reactions. In summary, although Ph-NO₂ and MOR have been studied previously, our study constructed a novel catalyst and proposed a unique mechanism. Our work makes valuable contributions to the electrochemical aniline synthesis and methanol electrooxidation to formate. Moreover, the design of this single-atom alloy metallene structure and the use of H*-spillover effect and synergistic catalytic effect are promising strategies for applying in other organic electrocatalytic conversion reactions. We believe that our study could provide some favourable insights into the design of catalysts for organic electrocatalytic conversion. Please page 19-20 and Ref. 37, 45, 56.

Revision made in manuscript: Please see Page 19-20 and Ref. 37, 45, 56. We have highlighted them in the marked-up revised manuscript.

The superior Ph-NO₂ ERR and MOR activity of Cu_{SA}-Rh MAs/CF originates from the following points: Firstly, the stable security wall-like structure formed by the ultrathin metallene arrays provides sufficient active sites and abundant interlayer channels⁴⁵. Secondly, the inherent defect-

rich structure and low-crystalline regions of Cu_{SA}-Rh MAs/CF can induce unsaturated coordination metallic bonds and optimize the local electron structure^{37,56}. Thirdly, the synergistic catalysis effect and H*-spillover effect between Cu single-atom and Rh host can optimize the catalytic reaction process, facilitate the stable and rapid conversion of reactants to intermediates as well as accelerate the desorption of target products. Fourthly, the Cu single-atom as effective adsorption sites can modulate the competition for adsorbate adsorption on Rh sites thus promoting electrocatalytic reactions.

37. Prabhu P & Lee JM. Metallenes as functional materials in electrocatalysis. *Chem. Soc. Rev.* **50**, 6700-6719 (2021).
45. Babar P, *et al.* Bifunctional 2D electrocatalysts of transition metal hydroxide nanosheet arrays for water splitting and urea electrolysis. *ACS Sustainable Chem. Eng.* **7**, 10035-10043 (2019).
56. Cao D, Xu H & Cheng D. Construction of defect-rich RhCu nanotubes with highly active Rh₃Cu₁ alloy phase for overall water splitting in all pH values. *Adv. Energy Mater.* **10**, 1903038 (2020).